# Impact of Connectivity on Laplacian Representations in Reinforcement Learning

**Tommaso Giorgi** [1]   **Pierriccardo Olivieri** [2]   **Keyue Jiang** [3]   **Laura Toni** [3]   **Matteo Papini** [4]

## Abstract

Learning compact state representations in Markov Decision Processes (MDPs) has proven crucial for addressing the curse of dimensionality in large-scale reinforcement learning (RL) problems. Existing principled approaches leverage structural priors on the MDP by constructing state representations as linear combinations of the state-graph Laplacian eigenvectors. When the transition graph is unknown or the state space is prohibitively large, the graph spectral features can be estimated directly via sample trajectories. In this work, we prove an upper bound on the approximation error of linear value function approximation under the learned spectral features. We show how this error scales with the algebraic connectivity of the state-graph, grounding the approximation quality in the topological structure of the MDP. We further bound the error introduced by the eigenvector estimation itself, leading to an end-to-end error decomposition across the representation learning pipeline. Additionally, we show how the common expression for the symmetrized MDP Laplacian is easy to misinterpret, and propose a more straightforward reformulation. Our results hold for general (non-uniform) policies without any assumptions on the symmetry of the induced transition kernel. We validate our theoretical findings with numerical simulations on gridworld environments.

## 1. Introduction

At the core of reinforcement learning (RL, Sutton & Barto, 2018), an agent aims to incrementally learn an optimal behavior through experience by interacting with an environment. Evaluating the agent's behavior, or policy, in terms of the expected cumulative reward is a fundamental problem in RL literature, known as policy evaluation. The objective of policy evaluation is to compute or estimate the value function induced by the agent's policy and the environment dynamics. Standard model-based methods for policy evaluation (Bertsekas, 2025) are often constrained by the size of the problem or by the specific task given to the agent, making exact solutions computationally intractable in high-dimensional or continuous state spaces. Value function approximation addresses this issue by mapping the value function to a low-dimensional feature space defined by a representation map $\psi$, trading bias for computational tractability. End-to-end approaches based on nonlinear function approximation, such as deep neural networks, implicitly learn this low-dimensional embedding of the state space (Mnih et al., 2015). In contrast, linear function approximation methods construct the value function as a linear combination of explicit state features given by a fixed map $\psi$. While end-to-end deep models are abundantly investigated from an empirical perspective, linear models often offer a better understanding of the foundations of RL with function approximation (Agarwal et al., 2019). Moreover, deep learning can still play a prominent role in the preliminary learning of the feature map.

Mapping the state transitions into a graph (nodes being the states and edges reflecting state transitions induced by a behavior policy) allows us to construct basis functions as the eigenvectors of the state-transition graph Laplacian (Mahadevan & Maggioni, 2007). The Laplacian basis is expressive, interpretable and inherits desirable properties: (i) it naturally captures the underlying topology of the environment as the extracted set of features explains hidden geometrical structures and symmetries of the MDP and (ii) it is not tied to any particular reward function. This latter property is of particular interest in the context of multi-task or unsupervised reinforcement learning, where an agent encounters different reward specifications within the same environment.

Under this representation, the problem boils down to (i) constructing the graph, and (ii) learning the node representation. In model-based approaches, the graph is constructed explicitly. Although this approach scales poorly with the size of the state space, most existing theoretical analyses focus on this. In this work we turn our attention to the

[1]Baker Hughes, Florence, Italy [2]Politecnico di Milano [3]University College London, AI Center (UCL) [4]Università degli Studi di Milano. Correspondence to: Pierriccardo Olivieri <pierriccardo.olivieri@polimi.it>.

*Proceedings of the $43^{rd}$ International Conference on Machine Learning*, Seoul, South Korea. PMLR 306, 2026. Copyright 2026 by the author(s).

model-free approach, where the representation is learned directly from interaction data without an explicit model of the transition graph. As proposed by Wu et al. (2019), this can be done via stochastic optimization, namely by optimizing the Graph Drawing Objective (GDO) introduced by Koren (2003). Subsequent works refined this objective in order to address some of its shortcomings (Wang et al., 2021), with the most recent (Gomez et al., 2024) proposing the Augmented Lagrangian Laplacian Objective (ALLO). Interested readers can refer to Section 5 for a comprehensive review of related works.

Understanding the error of the learned representation, i.e., the quality of the approximated value function for any given downstream task and policy, is fundamental for its application to policy evaluation (and possibly, optimization) with linear function approximation. Both the spectral decomposition and GDO optimization phases contribute to the approximation error of the Laplacian representation. To our knowledge, there are no characterizations of the overall approximation error, which is crucial to understand the limitations of the Laplacian representation and which factors contribute the most to the error. This may help practitioners selecting the number of features, the behavior policy used to collect the data, or anticipating failure modes in poorly connected MDPs. In the study of Laplacian representations for RL, it is customary to assume the symmetry of the induced transition matrix for simplicity (Mahadevan & Maggioni, 2007). This assumption is satisfied, for example, by deterministic decision processes with uniform policies. However, the symmetry assumption excludes most cases of practical interest. In this work, we make no assumptions on the symmetry of the dynamics, admitting general stochastic transitions and non-uniform behavior policies. In fact, while symmetry certainly helps, our analysis suggests that the quality of the approximation is fundamentally governed by the transition graph's *connectivity*.

**Our contribution.** In this work, we consider the average reward setting (Puterman, 2014) and decompose the approximation error of a learned Laplacian representation into two components:

1. The approximation error conditioned on the exact knowledge of the Laplacian. In particular, we provide an upper bound on the error introduced by the truncated spectral decomposition. We show that it scales with $\lambda_2$, the *spectral gap*, corresponding to the smallest nonzero eigenvalue of the Laplacian. We highlight the link between this error and the connectivity of the underlying graph.

2. The error induced by learning the graph features from data. In particular, we provide an upper bound on the additional approximation error originating from the

estimation of the Laplacian eigenvectors by means of the Graph Drawing Objective.

Additionally, we propose a reformulation of the Laplacian for general (non-symmetric) MDPs that prevents some possible misunderstandings, of which we show some examples from the literature. Finally, we empirically validate the impact of the MDP connectivity on the value function approximation error in simulated gridworld environments.

## 2. Preliminaries

**Notation.** For a vector $v$ and matrices $A, D$, $\|v\|_D = \sqrt{v^\top D v}$ denotes the weighted (by $D$) $\ell_2$ (Euclidean) norm and $\|A\|_D = \max_{v \neq 0} \|Av\|_D / \|v\|_D$ the corresponding weighted operator norm; $\delta_{i,j}$ is the Kronecker delta and is equal to $1$ if $i = j$, $0$ if $i \neq j$.

**Infinite-Horizon Average Reward MDP.** An infinite-horizon average-reward Markov Decision Process (MDP) is defined by the tuple $\langle \mathcal{S}, \mathcal{A}, \mathcal{P}, r \rangle$, where $\mathcal{S}$ is a finite set of states, $\mathcal{A}$ is a finite set of actions, $\mathcal{P}(\cdot|s, a)$ is a transition kernel over $\mathcal{S}$, $r : \mathcal{S} \times \mathcal{A} \to \mathbb{R}$ is the reward function, where $r(s, a)$ is the expected reward for taking action $a$ in state $s$. A policy $\pi$ is a mapping from states to probability distributions over actions, i.e. $\pi(a|s)$ denotes the probability of selecting action $a$ in state $s$. Given a policy $\pi$, actions are sampled as $a_t \sim \pi(\cdot|s_t)$, and the environment transitions according to $s_{t+1} \sim \mathcal{P}(\cdot|s_t, a_t)$. A policy $\pi$ induces a time-homogeneous Markov chain over the state space $\mathcal{S}$ with transition kernel $\mathcal{P}^\pi(s'|s) := \sum_{\mathcal{A}} \pi(a|s)\mathcal{P}(s'|s, a)$. We assume that, for any policy $\pi$, the induced Markov chain over $\mathcal{S}$ is irreducible and aperiodic, and thus admits a unique stationary distribution $\phi^\pi$. For a fixed policy $\pi$, define the long-run average reward from an initial state $s$ as

$$\rho_\pi(s) := \lim_{T \to \infty} \frac{1}{T} \mathbb{E}^\pi_{s_0=s} \left[ \sum_{t=0}^{T-1} r(s_t, a_t) \right].$$

Under the irreducibility and aperiodicity assumptions, $\rho_\pi(s)$ is independent of the initial state. We therefore denote this state-independent quantity by $\rho_\pi$, which can be written as

$$\rho_\pi = \sum_{s \in \mathcal{S}} \phi^\pi(s) \sum_{a \in \mathcal{A}} \pi(a|s)\, r(s, a),$$

where $\phi^\pi$ denotes the stationary distribution induced by $\pi$. We define the expected per-state reward $r_\pi(s)$ and the centered reward vector $\bar{r}_\pi$ as follows:

$$r_\pi(s) := \sum_{a \in \mathcal{A}} \pi(a|s)\, r(s, a), \quad \bar{r}_\pi := r_\pi - \rho_\pi \mathbf{1},$$

where $\mathbf{1}$ is the all-ones vector. The (differential) value function $v_\pi \in \mathbb{R}^{|\mathcal{S}|}$ is defined as any solution to the Poisson equation

$$v_\pi = \bar{r}_\pi + \mathcal{P}^\pi v_\pi.$$

Since $v_\pi$ is unique only up to an additive constant, we adopt the normalization (Sutton et al., 1999):

$$\langle v_\pi, \phi \rangle = v_\pi^\top \phi = 0,$$

which yields a unique $v_\pi$. Throughout the paper we fix an arbitrary policy $\pi$. For notational simplicity, we write $P$, $\phi$, $v$, $r$ and $\bar{r}$ in place of $\mathcal{P}^\pi$, $\phi^\pi$, $v_\pi$, $r_\pi$ and $\bar{r}_\pi$ respectively.

**The Graph Laplacian.** Given a weighted and undirected graph $G = (\mathcal{V}, \mathcal{E})$ with symmetric weight matrix $W$, the weighted combinatorial graph Laplacian of $\mathcal{G}$ is defined as:

$$L = D - W,$$

where $D$ is a diagonal matrix where the $i$-th diagonal entry is $D_{ii} = \sum_{j=1}^{|\mathcal{V}|} W_{ij}$. The spectrum of the Laplacian matrix encodes the geometrical properties of the graph, in particular its connectivity and the presence of bottlenecks. The smallest eigenvalue of $L$ is always zero, and its multiplicity equals the number of connected components of the graph. When the graph is strongly connected, the second smallest eigenvalue $\lambda_2$, often called *algebraic connectivity*, is linked to the Cheeger constant $h(G)$ of the graph (Chung, 1997), which provides a quantitative measure of how well connected the graph is. In particular, small values of $h(G)$ indicate the presence of sparse cuts that separate the graph, while larger values imply strong global connectivity. A relation between $h(G)$ and $\lambda_2$ is the Cheeger inequality (Chung, 1997):

$$\frac{h(G)^2}{2} \le \lambda_2(L) \le h(G). \tag{1}$$

Chung (2005) defines the Laplacian matrix of the graph induced by an ergodic Markov chain with transition probability matrix $P$ which is directed in general, as $P$ is not symmetric. Since the definition of the Laplacian requires the weight matrix to be symmetric, Chung (2005) defines $W$ as follows:

$$W = \frac{\Phi^{\frac{1}{2}} P \Phi^{-\frac{1}{2}} + \Phi^{-\frac{1}{2}} P^\top \Phi^{\frac{1}{2}}}{2}.$$

The corresponding normalized Laplacian is defined as:

$$\mathcal{L} = D - W = I - \frac{\Phi^{\frac{1}{2}} P \Phi^{-\frac{1}{2}} + \Phi^{-\frac{1}{2}} P^\top \Phi^{\frac{1}{2}}}{2}, \tag{2}$$

where $\Phi$ is the diagonal matrix where the $i$-th diagonal entry is $\Phi_{ii} = \phi_i$ and $\phi$ is the stationary distribution under $P$. Under this formulation, Chung (2005) shows an analog of the Cheeger inequality that we omit for brevity.[1]

---

[1] For directed graphs, $h(G)$ is defined using the notion of circulation (Chung, 2005).

**The Laplacian Representation in RL.** Among the different approaches to compute the feature map $\psi$ for a given MDP, Mahadevan & Maggioni (2007) propose a spectral framework, where $\psi : \mathcal{S} \to \mathbb{R}^k$ is obtained by taking the eigenvectors $(\psi_1, \ldots, \psi_k)$ associated with the lowest $k$ eigenvalues of the Laplacian of a graph $(\mathcal{S}, \mathcal{E})$ constructed as follows: the nodes are the states of the MDP, and the edges are the state transitions induced by the MDP dynamics $\mathcal{P}$ and the agent's behavior policy $\pi$, weighted by their probability. Namely, the weight $W_{ij}$ associated with edge $(s_i, s_j)$ is $\mathcal{P}^\pi(s_j|s_i) = \sum_{a \in \mathcal{A}} \pi(a|s_i) \mathcal{P}(s_j|s_i, a)$ or just $P(s_j|s_i)$ in our compact notation. The transition probability matrix $P$ is often assumed to be symmetric, and the Laplacian matrix reduces to:

$$L = I - P.$$

By the Courant-Fischer theorem (Horn & Johnson, 2012, theorem 4.2.11) the eigenbasis $(\psi_1, \ldots, \psi_k)$ minimizes:

$$\begin{cases} \min_{\{x_i\}_{i=1}^k} \sum_{i=1}^k x_i^\top L x_i, \\ x_i^\top x_j = \delta_{ij}, \quad 1 \le i \le j \le k. \end{cases} \tag{3}$$

For a Laplacian matrix, for any $x \in \mathbb{R}^{|\mathcal{V}|}$, its quadratic form takes the following convenient form:

$$x^\top L x = \sum_{i \in \mathcal{V}} \sum_{j \in \mathcal{V}} W_{ij} \big( x(i) - x(j) \big)^2 \tag{4}$$

which is related to the smoothness in the sense of 2-Dirichlet form (the smaller, the smoother $x$ is on the graph).[2] Given the recursive nature of the value function, the common assumption is that $v$ is smooth so that $\psi$ provides a suitable representation.

**Graph Drawing Objective (GDO).** The minimization problem from Equation (3) was introduced by Koren (2003) and is known as Graph Drawing Objective (GDO). By leveraging Equation (4), the GDO can be expressed as an expectation, making the optimization suitable for model-free reinforcement learning. In particular, the weights $W_{ij}$ can be interpreted as probabilities, according to which data (state pairs $i, j$) are sampled to minimize the mean squared error $\sum_{s_i, s_j \in \mathcal{S}} (x(s_i) - x(s_j))^2$.

Using this intuition, Wu et al. (2019), Wang et al. (2021) and Gomez et al. (2024) propose different optimization techniques. These approaches make use of deep learning and parameterize the spectral representation $\psi^\theta : \mathcal{S} \to \mathbb{R}^k$.

---

[2] The notion of smoothness of a function $f$ on a graph $\mathcal{G}$ can be expressed by the discrete $p$-Dirichlet form of $f$ on $\mathcal{G}$ (Shuman et al., 2013): $S_p(f) := \frac{1}{p} \sum_{i \in \mathcal{V}} \|\nabla_i f\|_2^p = \frac{1}{p} \sum_{i \in \mathcal{V}} \left[ \sum_{j \in \mathcal{V}} W_{ij} (f(j) - f(i))^2 \right]^{\frac{p}{2}}$.

Namely, given a state $s$, the network is trained to output a $k$-dimensional vector representing the coordinates of $s$ in the space spanned by the $k$ eigenvectors corresponding to the smallest eigenvalues. The objective function to be minimized is:

$$l(\theta) = \mathbb{E}_{s\sim\phi,\, s'\sim P(\cdot|s)}\left[\sum_{i=1}^{k}\left(\psi_i^\theta(s) - \psi_i^\theta(s')\right)^2\right] + \mathcal{R}(\theta),$$

where $\mathcal{R}$ is a regularizer intended to induce orthogonality, typically a relaxation of $\mathbb{E}_{s\sim\phi}\left[\psi_i^\theta(s)\psi_j^\theta(s) - \delta_{ij}\right] = 0$. For example, Wu et al. (2019) propose:

$$\mathcal{R}(\theta) = \beta\mathbb{E}_{s,s'\sim\rho}\left[\sum_{i=1}^{k}\sum_{j=1}^{k}(\psi_i^\theta(s)\psi_j^\theta(s) - \delta_{ij})\right.$$
$$\left.\cdot\, (\psi_i^\theta(s')\psi_j^\theta(s') - \delta_{ij})\right], \quad (5)$$

where $\beta > 0$ is a regularization hyperparameter.

## 3. Approximation Error

To the best of our knowledge, the graph induced by the state transitions is usually assumed to be undirected, implying P to be symmetric. This is a limiting assumption compared to empirical RL problems in which the graph induced by the MDP and the behavior policy is often far from symmetric. In contrast, we adopt the following definition of the Laplacian matrix:

$$L = I - \frac{P + \Phi^{-1}P^\top\Phi}{2}, \quad (6)$$

which is $\Phi$-self-adjoint even when $P$ is not symmetric. This allows us to employ the usual tools of graph spectral analysis without making strong assumptions on the transition dynamics. Moreover, our definition of $L$ reflects what is actually estimated in practice with GDO-based minimization procedures, as explained thoroughly in Section 4. This makes our study of the approximation error of the representation based on $L$ practically relevant. Additionally, a similarity relation holds between $L$ and the Laplacian matrix for directed graphs from Equation (2), allowing us to apply some of the topological intuition developed by Chung (2005).

Given the above definition of the Laplacian, in this section we derive an upper bound on the approximation error of the value function approximated as a linear function of the first $k$ eigenvectors of $L$. To be able to focus on the approximation error of the learned representation, we make the following assumptions, where we fix a policy and a dimension $k \leq |\mathcal{S}|$.

**Assumption 3.1** ($\epsilon$-optimal GDO). We have access to an algorithm (hereafter called GDO) that outputs features

$\widehat{u}_1, \dots \widehat{u}_k \in \mathbb{R}^{|\mathcal{S}|}$ such that $\widehat{u}_i^\top\Phi\widehat{u}_j = \delta_{ij}$ for all $i, j = 1, \dots, k$ and

$$\sum_{i=1}^{k}\widehat{u}_i^\top\Phi L\widehat{u}_i - \sum_{i=1}^{k}\lambda_i < \epsilon,$$

where $\lambda_1, \dots, \lambda_k$ are the $k$ smallest eigenvalues of the Laplacian $L$ from Equation 6.

We will discuss in Section 4 why this is a reasonable characterization of the residual error $\epsilon$ of common Laplacian representation learning algorithms.

**Assumption 3.2** (Linear least-squares oracle). We have access to a $\Phi$-weighted linear least-squares oracle that, given features $x_1, \dots x_k \in \mathbb{R}^{|\mathcal{S}|}$ (not necessarily orthonormal), returns $\widehat{v}_k := \Pi_X^\Phi v$, where $X$ is the matrix having $x_1, \dots x_k$ as columns and $\Pi_X^\Phi = X(X^\top\Phi X)^{-1}X^\top\Phi$ is the $\Phi$-weighted projection onto its column space.

This latter assumption allows us to ignore the estimation error of the coefficients of the linear value function and focus on the approximation error. The weighting by $\Phi$ reflects the realistic scenario where the coefficients are learned from on-policy data.

### 3.1. Main Theorem

We consider the learned $k$-dimensional representation $\widehat{\psi}(s) = (\widehat{u}_1(s), \dots, \widehat{u}_k(s))$ obtained by minimizing GDO, assuming perfect orthonormality and an $\epsilon$-optimality gap.

**Theorem 3.3.** *Let $u_1, \dots, u_{|\mathcal{S}|}$ be the eigenvectors of $L$ associated with eigenvalues $\lambda_1 \leq \dots \leq \lambda_{|\mathcal{S}|}$. Denote as $\widehat{u}_1, \dots, \widehat{u}_k$ the features produced by $\epsilon$-optimal GDO (Assumption 3.1) and $\widehat{v}_k$ the approximation of $v$ produced by the $\Phi$-weighted linear least-squares oracle (Assumption 3.2) using these features. Then:*

$$\|v - \widehat{v}_k\|_\Phi \leq \|\bar{r}\|_\Phi\sqrt{\frac{1}{\lambda_2\lambda_{k+1}}} + \|v\|_\Phi\sqrt{\frac{2\epsilon}{\lambda_{k+1} - \lambda_k}}.$$

The bound consists of two terms: (i) an error arising from the truncation up to $k < |\mathcal{S}|$ eigenvectors, denoted as *truncation error* and (ii) an error accounting for the representation learning procedure, denoted as *feature estimation error* in the following. The former:

$$\|\bar{r}\|_\Phi\sqrt{\frac{1}{\lambda_2\lambda_{k+1}}},$$

highlights an explicit dependence on the spectral gap $\lambda_2$. We note that given the irreducibility assumption, the resulting graph is connected thus $\lambda_2$ is strictly greater than 0. The latter:

$$\|v\|_\Phi\sqrt{\frac{2\epsilon}{\lambda_{k+1} - \lambda_k}}$$

scales with the residual error $\epsilon$ of GDO and depends on the gap between the largest included and the smallest discarded eigenvalues of $L$. The two terms are derived separately in the following subsections.

## 3.2. Truncation Error

Using a truncated Laplacian eigenbasis yields a low-dimensional representation that makes policy estimation feasible in large-scale environments. However, this dimensionality reduction introduces a source of error, the truncation error.

We provide a bound on this quantity described by the following lemma:

**Lemma 3.4.** *Let $v_k$ be the approximation of $v$ produced by the $\Phi$-weighted linear least-squares oracle (Assumption 3.2) using the (exact) $k$ eigenvectors of $L$ associated with the $k$ smallest eigenvalues. Then:*

$$\|v - v_k\|_\Phi^2 \leq \frac{\|\bar{r}\|_\Phi^2}{\lambda_2 \lambda_{k+1}}$$

*where $0 = \lambda_1 \leq \lambda_2 \leq \ldots \leq \lambda_{|\mathcal{S}|}$.*

We note that, in contrast with Theorem 3.3, here we bound the error with respect to $v_k$ which is the approximation using the true Laplacian eigenvectors. The bounding factor depends on the spectral gap $\lambda_2$, also known as Fiedler eigenvalue or algebraic connectivity, and becomes smaller as we increase the dimensionality of the representation. The dependence on $\lambda_2$ shows how well-connected MDPs provide representations which yield lower error and can be interpreted in light of the Cheeger inequality (Eq. 1). The numerator captures the magnitude of the centered reward vector under the stationary distribution. In particular, policies that concentrate probability mass on states where the reward exhibits large deviations from its mean lead to larger error.

## 3.3. Estimation Error

GDO optimization provides a model-free way to estimate the Laplacian eigenvectors associated with the $k$ lowest eigenvalues directly from interaction data. This estimation introduces a source of error, the estimation error. Linear function approximation relies on orthogonal projection operators onto a certain basis. We prove a bound on the spectral norm of the difference between two linear operators: $\Pi_\Psi$ the projector onto the exact eigenbasis and (ii) $\Pi_{\tilde{\Psi}}$ the projector onto the approximated eigenbasis. We first derive a general result that can be applied to any Laplacian operator:

**Lemma 3.5.** *[Graph Drawing Lemma] Let $u_1, \ldots, u_{|\mathcal{S}|}$ be the eigenvectors of a p.s.d. symmetric matrix $A$ associated with eigenvalues $0 = \lambda_1(A) < \lambda_2(A) \leq \cdots \leq \lambda_{|\mathcal{S}|}(A)$.*

*Let $\tilde{u}_1, \ldots, \tilde{u}_k \in \mathbb{R}^{|\mathcal{S}|}$ be orthonormal vectors satisfying $\tilde{u}_i^\top \tilde{u}_j = \delta_{ij}$ for all $i, j \in \{1, \ldots, k\}$.*

*Define $\Psi := (u_1, \ldots, u_k) \in \mathbb{R}^{|\mathcal{S}| \times k}$ and $\tilde{\Psi} := (\tilde{u}_1, \ldots, \tilde{u}_k) \in \mathbb{R}^{|\mathcal{S}| \times k}$, and let $\Pi_\Psi$ and $\Pi_{\tilde{\Psi}}$ denote the orthogonal projection matrices onto $\mathrm{span}(\Psi)$ and $\mathrm{span}(\tilde{\Psi})$, respectively.*

*Suppose there exists $\epsilon > 0$ such that*

$$\sum_{i=1}^k \tilde{u}_i^\top A \tilde{u}_i - \sum_{i=1}^k \lambda_i < \epsilon.$$

*Then,*

$$\|\Pi_\Psi - \Pi_{\tilde{\Psi}}\|_2 < \sqrt{\frac{2\epsilon}{\lambda_{k+1}(A) - \lambda_k(A)}}.$$

The result follows from a Davis–Kahan-type sin$\Theta$ argument (Davis & Kahan, 1970) and shows a dependence on the gap between $\lambda_{k+1}$ (the first discarded eigenvalue) and $\lambda_k$. Although our result does not provide theoretical guarantees that this difference is strictly positive, a recent work Christoffersen et al. (2024) shows that with high probability this gap is nonzero for the Laplacian matrix of random Erdős-Rényi graphs. The case where this difference is zero signals the presence of an eigenvalue with multiplicity strictly greater than one which makes the spectral description structurally incomplete and causes the bound to diverge. Nevertheless, we provide theoretical guarantees (Corollary A.1) for this scenario together with an extended discussion on the matter in Appendix A. Additionally $\lambda_k$ is related to the $k$-way expansion constant of the graph which measures the cost of the best partition of the nodes into $k$ pieces via a Cheeger-like inequality (Lee et al., 2014). Small values of $\lambda_k$ are associated with a graph that can be cheaply cut into $k$ separated groups, therefore approximately $k$ disconnected clusters. The eigengap $\lambda_{k+1} - \lambda_k$ then quantifies how well the graph decomposes into $k$ clusters as opposed to $k + 1$.

We can apply the Graph Drawing Lemma to our case as follows.

**Lemma 3.6.** *Let $u_1, \ldots, u_{|\mathcal{S}|}$ be the eigenvectors of $L$ associated with eigenvalues $0 = \lambda_1 \leq \lambda_2 \leq \cdots \leq \lambda_{|\mathcal{S}|}$. Let $\widehat{u}_1, \ldots, \widehat{u}_k \in \mathbb{R}^{|\mathcal{S}|}$ be their estimates produced by $\epsilon$-optimal GDO (Assumption 3.1). Define $\Psi := (u_1, \ldots, u_k) \in \mathbb{R}^{|\mathcal{S}| \times k}$ and $\widehat{\Psi} := (\widehat{u}_1, \ldots, \widehat{u}_k) \in \mathbb{R}^{|\mathcal{S}| \times k}$. Then,*

$$\|\Pi_\Psi^\Phi - \Pi_{\widehat{\Psi}}^\Phi\|_\Phi < \sqrt{\frac{2\epsilon}{\lambda_{k+1} - \lambda_k}}.$$

The upper bound captures the error introduced by using $\widehat{u}_1, \ldots \widehat{u}_k$ instead of $u_1, \ldots, u_k$ in representing the value function, in terms of the residual error $\epsilon$ of GDO and the

gap between the smallest discarded and the largest included eigenvalues of $L$, which is typically nonzero for the reasons discussed above.

To the best of our knowledge, the results presented in this section are the first theoretical guarantees on the accuracy of GDO.

# 4. Remarks on the Definition of the Laplacian

Wu et al. (2019) provided a very general definition of the Laplacian operator for RL. They only assumed the state space $\mathcal{S}$ and the stationary distribution of interest $\phi$ to form a measure space, a setting that accommodates discrete and continuous MDPs. They developed their theory in the Hilbert space $\widetilde{\mathcal{H}}$ of $\phi$-square-integrable functions, and defined the Laplacian as a Hilbert-Schmidt integral operator. When $\mathcal{S}$ is finite, say $|\mathcal{S}| = d$, $\widetilde{\mathcal{H}}$ is $\mathbb{R}^d$ equipped with the inner product:

$$\langle f, g \rangle_{\widetilde{\mathcal{H}}} = \sum_{s \in \mathcal{S}} f(s)g(s)\phi(s), \tag{7}$$

and any linear operator $\widetilde{K} : \widetilde{\mathcal{H}} \to \widetilde{\mathcal{H}}$ acts on $f \in \widetilde{\mathcal{H}}$ as follows:

$$\widetilde{K}f(s) = \sum_{s' \in \mathcal{S}} \widetilde{K}(s, s')f(s')\phi(s'), \tag{8}$$

where $\widetilde{K} : \mathcal{S} \times \mathcal{S} \to \mathbb{R}$ (overloading the notation) is their kernel function. We mark these operators with a tilde to emphasize they are *not* linear operators on $\mathbb{R}^d$.[3] Indeed, the incautious reader of (Wu et al., 2019) may overlook this subtle but important aspect and interpret inner products as Euclidean dot products and linear operators as matrices—that is, essentially missing the weighting by $\phi$ in dot products. For example, Touati et al. (2023) and Gomez et al. (2024) report expressions for the general Laplacian that are incorrect unless one makes strong assumptions on the transition matrix (see App. B for further details). Fortunately, these oversights do not affect their main results (these works mainly focus on the symmetric case anyway, and Gomez et al. (2024) include a correct general formulation in the Appendix), but they serve as examples of how the abstract formulation can be misleading.

To avoid any further confusion, let us go back to the Laplacian operator as originally defined by Wu et al. (2019):[4]

$$\widetilde{L}f = (\widetilde{I} - \widetilde{D})f, \tag{9}$$

where

$$\widetilde{D}(s, s') = \frac{P(s'|s)}{2\phi(s')} + \frac{P(s|s')}{2\phi(s)}. \tag{10}$$

---

[3]They could be interpreted as such by incorporating $\phi$ in their kernel function, but this would be misleading.

[4]Note that the identity $\widetilde{I}$ of $\widetilde{\mathcal{H}}$ is also not the same as the identity $I$ of $\mathbb{R}^d$.

We argue that our expression for the Laplacian (Eq. 6), although less general, is much more usable, as it is designed for Euclidean spaces with the canonical notions of inner product and linear operators.

The equivalence between our definition and that of Wu et al. (2019) is established by the following:[5]

**Proposition 4.1.** *When $\mathcal{S}$ is finite, for any pair of functions $f, g : \mathcal{S} \to \mathbb{R}$,*

$$\langle f, g \rangle_{\widetilde{\mathcal{H}}} = f^\top \Phi g \quad and \quad \langle f, \widetilde{L}g \rangle_{\widetilde{\mathcal{H}}} = f^\top \Phi L g,$$

*where $L$ is defined in Equation (6).*

In Appendix B we prove a more general version of Proposition 4.1 that holds under a generic reference measure, accommodating both discrete (not necessarily finite) and continuous state spaces.

We have established an equivalence of sort between $\widetilde{L}$ and $L$. For our purposes, the scope of this equivalence lies entirely in the computation and estimation of Laplacian eigenvectors. Continuing to follow Wu et al. (2019), we lay down the following optimization problem

$$\min_{x_1,\ldots,x_k} \quad \sum_{i=1}^{k} \langle x_i, \widetilde{L}x_i \rangle_{\widetilde{\mathcal{H}}} \tag{11}$$
$$\text{s.t.} \quad \langle x_i, x_j \rangle_{\widetilde{\mathcal{H}}} = \delta_{ij}, \quad i, j \in [k],$$

which Proposition 4.1 allows us to rewrite as

$$\min_{x_1,\ldots,x_k} \quad \sum_{i=1}^{k} x_i^\top \Phi L x_i \tag{12}$$
$$\text{s.t.} \quad x_i^\top \Phi x_j = \delta_{ij}, \quad i, j \in [k],$$

Applying a change of variables and the notion of matrix similarity (see App. B for details), one obtains the following:

**Proposition 4.2.** *The optimal value of Program (12) is $\sum_{i=1}^{k} \lambda_i$, where $\lambda_1, \ldots, \lambda_k$ are the $k$ smallest eigenvalues of $L$. Moreover, the solutions $x_1^*, \ldots, x_k^*$ are the first $k$ eigenvectors of $L$ according to the same ordering.*

This means that one can forget about the eigenfunctions of $\widetilde{L}$ in $\widetilde{\mathcal{H}}$ and just find the eigenvectors of the matrix $L$ from Equation (6). The obtained representation is the same. However, Program (12) also allows to re-derive the GDO. From Equation (4) and the properties of $\phi$, we obtain the following program, equivalent to the previous two:

$$\min_{x_1,\ldots,x_k} \quad \mathbb{E}_{s \sim \phi, s' \sim P(\cdot|s)} \left[ \sum_{i=1}^{k} (x_i(s) - x_i(s'))^2 \right]$$
$$\text{s.t.} \quad \mathbb{E}_{s \sim \phi} \left[ x_i(s)x_j(s) - \delta_{ij} \right] = 0, \quad i, j \in [k],$$

---

[5]The results of this section are just a revisitation of (Wu et al., 2019) under our reformulation of the Laplacian.

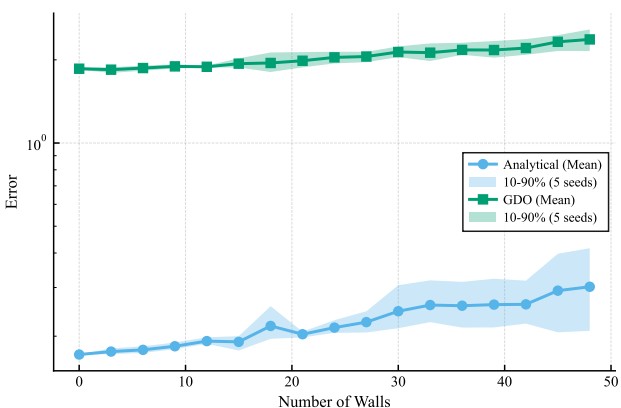

*Figure 1.* Shows the error between the true value function $v$ and approximate one. Two baselines are considered: the analytical case where the representation is obtained truncating the exact eigenvectors and its approximation obtained by optimizing GDO.

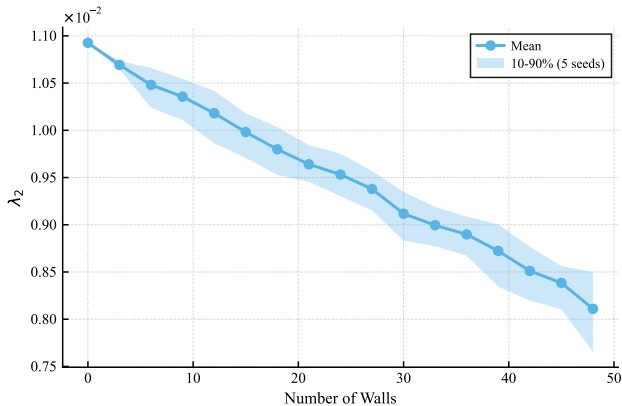

*Figure 2.* On the y-axis shows the log scale value assumed by $\lambda_2$, when the number of obstacles (walls) increases, or equivalently decreases the connectivity of the graph.

which is indeed the starting point of the GDO by Wu et al. (2019) and its subsequent developments. This, together with Proposition 4.2, also confirms that the error $\epsilon$ from Theorem 3.3 is the correct characterization of the residual error of GDO.

## 5. Related Works

Motivated by challenges such as generalization, sample efficiency, and scalability, representation learning has been widely used in reinforcement learning (RL), as auxiliary representations can compress large state spaces and mitigate the curse of dimensionality (Agarwal et al., 2020) to effectively scale RL methods to real-world applications. This facilitates the generality of RL in various downstream tasks. Along this line, many existing techniques lean on self-supervised or supervised learning. For instance, Oord et al. (2018) have adapted contrastive learning techniques to RL settings (Laskin et al., 2020; Eysenbach et al., 2022), where they learn the representation by distinguishing positive and negative samples to guide learning. Other methods may instead use an implicit representation through end-to-end methods to directly estimate the value function (Mnih et al., 2015; Chung et al., 2019).

To further improve efficiency and scalability, incorporating prior structure into RL to obtain low-dimensional representations has attracted huge research efforts (Mohan et al., 2024). One of the most promising directions is to utilize the spectral properties of the Markov Decision Process (MDP) for dimension reduction, particularly through Laplacian or the successor representation. Early work (Mahadevan & Maggioni, 2007; Osentoski & Mahadevan, 2007) developed the spectral framework in a model-based context by extracting representations from the leading eigenvectors of the MDP's Laplacian matrix. Recent work (Wu et al., 2019) further

extends the framework to the scenarios when structure is not available by formulating a graph drawing problem (Koren, 2003) for minimizing the Dirichlet energy, enabling model-free learning through stochastic optimization techniques. Subsequent works have refined the original Graph Drawing Objective (GDO) algorithm to address issues such as rotational ambiguity (Wang et al., 2021) and hyperparameter tuning of the GDO barrier (Gomez et al., 2024). A simplified objective was recently proposed by Ahmed et al. (2025) for online representation learning.

The spectral framework is appealing in part because it captures the environment topology while remaining independent of the reward function, which supports transfer across tasks. Its approximation properties have also been studied in Petrik (2007) who analyzed approximation error in the original proto-value function framework, and in Lan et al. (2022) who studied the statistical error of these methods in policy evaluation. Wang et al. (2022) highlighted a mismatch between Laplacian distances and state reachability and introduced a reachability-aware Laplacian representation rescaling with the eigenvalues. Spectral structure has also been extensively exploited in the bandit literature. For instance, spectral bandits (Valko et al., 2014) define arms as nodes in a graph and the reward function is assumed to be smooth w.r.t the graph Laplacian (e.g., small Dirichlet energy). Algorithms leverage lower-frequency Laplacian eigenvectors to effectively guide exploration and obtain regret bounds that depend on spectral quantities such as the Laplacian eigenvalue decay and effective dimension (Srinivas et al., 2010; Alon et al., 2017). This perspective is closely aligned with Laplacian representation learning in RL, in that low-frequency eigencomponents capture the intrinsic geometry and bottlenecks of the underlying state graph. A complementary line concerns spectrally low-rank bandits, where the expected reward depends on an unknown low-dimensional subspace that must be learned online (Jun

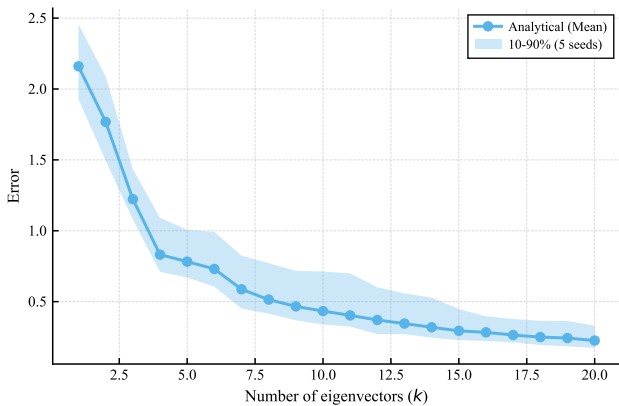

*Figure 3.* Shows the error of the analytical representation varying the number of eigenvectors $k$. On the y-axis we display the log scale value of the error, while on the x-axis the cut index $k$.

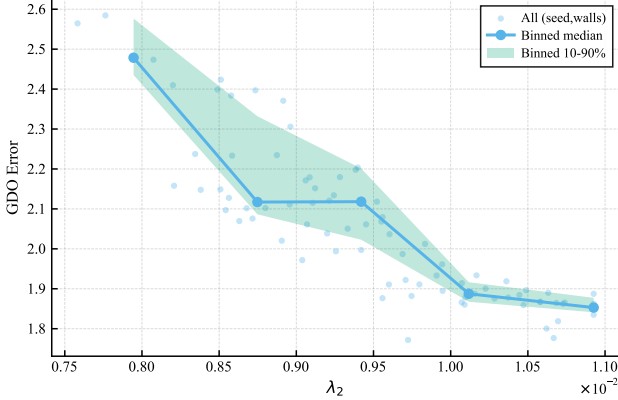

*Figure 4.* Shows the relationship between the second eigenvalue $\lambda_2$ and the error using the approximated representation via GDO. On the y-axis we display the value of the error, while on the x-axis the values assumed by $\lambda_2$, both metrics vary with respect to the number of walls considered.

et al., 2019). The algorithm aims to recover a principal subspace and then run bandit methods in the learned representation (Flynn et al., 2023; Auer et al., 2002). While these settings differ from MDP policy evaluation, they share the same idea of our analysis.

Other related work includes Proto-Value Networks, which adapt the proto-value function framework to deep architectures (Farebrother et al., 2023), and methods that learn spectral representations from dynamics without relying on the data collection policy (Ren et al., 2023). Finally, Laplacian-induced similarities derived from random walks have been explored for measuring node similarity in recommendation tasks (Fouss et al., 2007).

## 6. Numerical Simulations

The scope of this numerical evaluation is to validate and visualize the implications of the theoretical results derived in previous sections.[6] In particular, we want to highlight the role of connectivity, given by $\lambda_2$, in the truncation error. Visualizations of the impact of the connectivity of the MDP on $\lambda_2$ are in Appendix D, as well as additional simulations on stochastic environments and different tasks in Appendix E.

**Environment.** We consider a family of deterministic grid-world environments parameterized by the grid size, shaped by $n$ rows and $m$ columns and the number of walls $w$ with $n, m, w \in \mathbb{N}$. Each grid contains exactly a single goal located at the bottom-right corner for simplicity. The state space consists of all non-wall cells in the grid. The agent has access to four deterministic actions corresponding to the cardinal points. Transitions are deterministic, and if an action would lead the agent into a wall or out of bounds, the

---
[6]Code available at: https://github.com/pierriccardo/laprep

agent remains in its current state. The agent is rewarded $+1$ for transitioning to the goal state and receives $-1$ reward for all the other transitions. At reset, after reaching the maximum number of steps or the goal, the agent is teleported to a random state uniformly.

**Experimental Setup.** For each $w$-walls environment, we compute the true value function $v$ under a uniform policy using the average-reward Bellman equation with normalization $\phi^T v = 0$. Then, we obtain the representation considering the first $k$ eigenvectors of the Laplacian. This is obtained both via the analytical solution, and approximated by optimizing GDO. For both cases, we derive the linear approximation of the value function and compare it with the exact one. The weights are computed with the exact projectors to filter out the estimation error of least squares and focus on the approximation error of the representation. Finally, we test the ability of Laplacian-based representations to approximate value functions when connectivity varies. The aim is to empirically show that lower connectivity (higher $w$, lower $\lambda_2$) leads to more challenging approximation problems. To highlight the effects of connectivity, we simulate a series of environments with decreasing connections between states. This is achieved by progressively adding more walls to partially obstruct the path in the graph. Starting from an obstacle-free graph, we increase the number of walls $w$, by randomly removing an edge of the transition graph. We use a random spanning tree procedure to ensure that the graph remains connected.

**Results** The experimental evaluation aligns with our theoretical results. For this experiment, we fixed $n = m = 15$ and let the number of walls $w = \{1, \ldots, 50\}$ vary over 5 seeds. We cut the Laplacian eigenvectors at $k = 20$ for both the analytical solution and GDO. The plot shown in

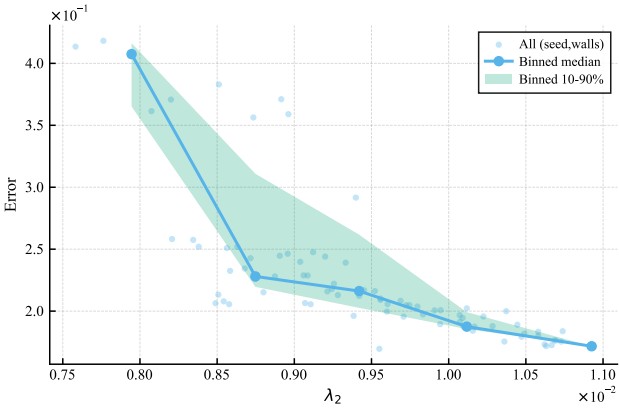

*Figure 5.* Shows the relationship between the second eigenvalue $\lambda_2$ and the error using the analytical eigenvector representation. On the y-axis we display the value of the error, while on the x-axis the values assumed by $\lambda_2$, both metrics vary with respect to the number of walls considered.

Figure 2 is used as a reference to visualize the decrease in the connectivity of the environment as the number of walls increases. The error comparison between the exact analytical derivation of the Laplacian eigenbasis and the one approximated by GDO is reported in Figure 1. The error is computed by comparing the $\Phi$-norm between the true value function $v$ and the one obtained using the representation. We denote as $v_k$ the approximation of $v$ using $k$ eigenvectors computed analytically. In the case of the analytical derivation of the basis, we use $\|v - v_k\|_\Phi$ where $k$ is the number of truncated eigenvectors. Similarly for GDO, we compute $\|v - \widehat{v}_k\|_\Phi$. Here, the two curves differ by the approximation error gap. However, as expected, they show the same increasing trend as the number of obstacles grows. Figure 4 and Figure 5 capture the influence of the connectivity $\lambda_2$ on the error using GDO-approximated and the true Laplacian eigenbasis, respectively, empirically confirming that reducing the connectivity increases the error. Finally, Figure 3 reports the error of the analytical representation when varying the cutoff dimension $k$, showing the expected decreasing behavior.

## 7. Conclusion

In this work, we provided quantifiable theoretical guarantees for the approximation error in the average reward setting using a Laplacian-based representation. We link this error to topological quantities related to the connectivity of the environment making the result interpretable. Finally we proposed a more straightforward reformulation of the Laplacian for non-symmetric transition graphs and validated the results in empirical settings. In future work, our theoretical analysis could be extended in two directions. Removing Assumption 3.1, one could study the sample complexity of a specific version of model-free GDO, that is, how the number of interaction steps required to achieve $\epsilon$-optimality scales with $\epsilon$. This is a question of (reward-free) representation learning. On the other end, removing Assumption 3.2, one could study the estimation error of linear least-squares for the learned representation. Results from (Lan et al., 2022) may prove useful. On the practical side, our upper bounds on the approximation error may be used to guide the choice of exploration policy to be used for representation learning and the dimension $k \ll |\mathcal{S}|$ of the representation. They may also guide the design of new representation algorithms based on the graph Laplacian.

## Impact Statement

This paper presents theoretical results and has the goal of advancing the field of Machine Learning. There are no potential societal consequences of our work which we feel must be highlighted here.

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

## A. Proofs and Further Discussion of Section 3

**Theorem 3.3.** *Let $u_1, \ldots, u_{|\mathcal{S}|}$ be the eigenvectors of $L$ associated with eigenvalues $\lambda_1 \leq \ldots \leq \lambda_{|\mathcal{S}|}$. Denote as $\widehat{u}_1, \ldots, \widehat{u}_k$ the features produced by $\epsilon$-optimal GDO (Assumption 3.1) and $\widehat{v}_k$ the approximation of $v$ produced by the $\Phi$-weighted linear least-squares oracle (Assumption 3.2) using these features. Then:*

$$\|v - \widehat{v}_k\|_\Phi \leq \|\bar{r}\|_\Phi \sqrt{\frac{1}{\lambda_2 \lambda_{k+1}}} + \|v\|_\Phi \sqrt{\frac{2\epsilon}{\lambda_{k+1} - \lambda_k}}.$$

*Proof.* Let $\Psi = (u_1, \ldots, u_k)$, the matrix whose columns are the eigenvectors of $L$ and $\widehat{\Psi} = (\widehat{u}_1, \ldots, \widehat{u}_k)$, define the $\Phi$-weighted projector onto $\Psi$, $\Pi_\Psi^\Phi$ as:

$$\Pi_\Psi^\Phi = \Psi(\Psi^\top \Phi \Psi)^{-1} \Psi^\top \Phi,$$

and the weighted projector onto $\widehat{\Psi}$, $\Pi_{\widehat{\Psi}}^\Phi = \widehat{\Psi}\widehat{\Psi}^\top \Phi$. Let $v_k$ be the linear function approximation of $v$ using $(u_1, \ldots, u_k)$, then by definition, $v_k = \Pi_\Psi^\Phi v$ and $\widehat{v}_k = \Pi_{\widehat{\Psi}}^\Phi v$, hence:

$$\|v - \widehat{v}_k\|_\Phi \leq \|v - v_k\|_\Phi + \|v_k - \widehat{v}_k\|_\Phi = \|v - v_k\|_\Phi + \|(\Pi_\Psi^\Phi - \Pi_{\widehat{\Psi}}^\Phi)v\|_\Phi$$

$$\|v - \widehat{v}_k\|_\Phi \leq \|v - v_k\|_\Phi + \|\Pi_\Psi^\Phi - \Pi_{\widehat{\Psi}}^\Phi\|_\Phi \|v\|_\Phi.$$

Now by Lemma 3.4 and Lemma 3.6, the thesis. $\qquad\square$

The main reason why a null eigengap is problematic for GDO-based objectives is rooted into the formulation of GDO itself. In particular GDO generally enforces conditions such as:

$$\sum_{i=1}^k \langle \widehat{u}_i, L\widehat{u}_i \rangle_\mathcal{H} - \sum_{i=1}^k \lambda_i < \epsilon$$

When the truncation at $k$ happens inside an eigenspace with dimension $b > 2$, we have that $\lambda_k = \ldots = \lambda_b$ and the objective function cannot distinguish directions inside the degenerate block, namely:

$$\sum_{i=1}^{k-1} \lambda_i + \lambda_k = \sum_{i=1}^{k-1} \lambda_i + \lambda_{k+1} = \ldots = \sum_{i=1}^{k-1} \lambda_i + \lambda_b$$

Therefore, any choice of a unit vector $v \in \text{span}(u_k, \ldots, u_b)$ yields the same objective value. In particular, for any such $v$,

$$\sum_{i=1}^{k-1} \lambda_i + \langle v, Lv \rangle_\mathcal{H} = \sum_{i=1}^{k-1} \lambda_i + \lambda_k,$$

so that replacing $u_k$ with $v$ produces another minimizer of the objective.

As a consequence, the optimizer is not unique: there exists a continuum of optimal $k$-dimensional subspaces of the form

$$\tilde{\Psi} = \text{span}(u_1, \ldots, u_{k-1}, v), \qquad v \in \text{span}(u_k, \ldots, u_b), \|v\| = 1.$$

But these subspaces can be arbitrarily far apart in operator norm for different choices of $v, w \in \text{span}(u_k, \ldots, u_b)$. Indeed when $v \perp w$:

$$\tilde{\Psi}_v = \text{span}(u_1, \ldots, u_{k-1}, v), \quad \tilde{\Psi}_w = \text{span}(u_1, \ldots, u_{k-1}, w),$$

both achieve the same objective value, yet

$$\|\Pi_{\tilde{\Psi}_v} - \Pi_{\tilde{\Psi}_w}\|_2 = 1.$$

To get theoretical guarantees in the null eigengap case, one has to ignore the extra representational power that $u_{k+1}, \ldots, u_b$ may provide. This is formalized in Corollary A.1.

**Corollary A.1.** *Let $u_1, \ldots, u_{|\mathcal{S}|}$ be the eigenvectors of $L$ associated with eigenvalues*

$$0 = \lambda_1 < \lambda_2 \leq \cdots \leq \lambda_h < \lambda_{h+1} = \cdots = \lambda_k = \lambda_{k+1} = \cdots = \lambda_b < \lambda_{b+1} \leq \cdots \leq \lambda_{|\mathcal{S}|}.$$

*In particular, $\lambda_{h+1}$ has multiplicity $b - h$, with eigenspace $\operatorname{span}(u_{h+1}, \ldots, u_b)$. Let $\widehat{u}_1, \ldots, \widehat{u}_k$ be the features produced by $\epsilon$-optimal GDO, and let $\widehat{v}_k$ be the approximation of $v$ produced by the $\Phi$-weighted linear least-squares oracle using these features. Then*

$$\|v - \widehat{v}_k\|_\Phi \leq \|\bar{r}\|_\Phi \sqrt{\frac{1}{\lambda_2 \lambda_{h+1}}} + \|v\|_\Phi \sqrt{\frac{2\epsilon}{\lambda_{h+1} - \lambda_h}}.$$

*Proof.* Let

$$H := \operatorname{span}(u_1, \ldots, u_h), \qquad \widehat{S} := \operatorname{span}(\widehat{u}_1, \ldots, \widehat{u}_k).$$

The main point is that, when $\lambda_k = \lambda_{k+1}$, the GDO objective does not identify the ordered prefix $\operatorname{span}(\widehat{u}_1, \ldots, \widehat{u}_h)$. Therefore we will not use that subspace. Instead, we first show that $\widehat{S}$ contains an $h$-dimensional subspace close to $H$.

It is convenient to pass to Euclidean coordinates. Define

$$A := \Phi^{1/2} L \Phi^{-1/2}, \qquad e_i := \Phi^{1/2} u_i, \qquad \widehat{e}_i := \Phi^{1/2} \widehat{u}_i.$$

$A$ is symmetric as proven in Lemma 3.4 (denoted as $\mathcal{L}$), the vectors $e_i$ form an orthonormal eigenbasis of $A$ with eigenvalues $\lambda_i$, and $\widehat{e}_1, \ldots, \widehat{e}_k$ are orthonormal. Let

$$\widehat{E} := \operatorname{span}(\widehat{e}_1, \ldots, \widehat{e}_k), \qquad E := \operatorname{span}(e_1, \ldots, e_h),$$

and let $\Pi_{\widehat{E}}$ be the Euclidean orthogonal projector onto $\widehat{E}$. By the $\epsilon$-optimality of GDO,

$$\operatorname{tr}(A \Pi_{\widehat{E}}) - \sum_{i=1}^k \lambda_i = \operatorname{tr}\left( \Phi^{\frac{1}{2}} L \Phi^{-\frac{1}{2}} \sum_{i=1}^k \Phi^{\frac{1}{2}} \widehat{u}_i \widehat{u}_i^\top \Phi^{\frac{1}{2}} \right) - \sum_{i=1}^k \lambda_i = \sum_{i=1}^k \widehat{u}_i^\top \Phi L \widehat{u}_i - \sum_{i=1}^k \lambda_i < \epsilon.$$

Let $\mu := \lambda_{h+1}$, so that $\lambda_{h+1} = \cdots = \lambda_b = \mu$. Since $\dim(\widehat{E}) = k$ and $(e_i)_{i=1}^{|\mathcal{S}|}$ is an orthonormal basis,

$$\sum_{i=1}^{|\mathcal{S}|} \langle e_i, \Pi_{\widehat{E}} e_i \rangle = \operatorname{tr}(\Pi_{\widehat{E}}) = k.$$

For compactness, write

$$\alpha_i := \langle e_i, \Pi_{\widehat{E}} e_i \rangle.$$

Then $\sum_{i=1}^{|\mathcal{S}|} \alpha_i = k$. Moreover, since $(e_i)_{i=1}^{|\mathcal{S}|}$ is an eigenbasis of $A$, we have

$$\operatorname{tr}(A \Pi_{\widehat{E}}) = \sum_{i=1}^{|\mathcal{S}|} \langle e_i, A \Pi_{\widehat{E}} e_i \rangle = \sum_{i=1}^{|\mathcal{S}|} \lambda_i \alpha_i.$$

Also, by the definition of $\mu$ and by the fact that $\lambda_{h+1} = \cdots = \lambda_k = \mu$,

$$\sum_{i=1}^k \lambda_i = \sum_{i=1}^h \lambda_i + (k - h)\mu.$$

Therefore

$$\operatorname{tr}(A \Pi_{\widehat{E}}) - \sum_{i=1}^k \lambda_i = \sum_{i=1}^{|\mathcal{S}|} \lambda_i \alpha_i - \left( \sum_{i=1}^h \lambda_i + (k - h)\mu \right)$$

$$= \sum_{i=1}^h \lambda_i (\alpha_i - 1) + \sum_{i=h+1}^b \mu \alpha_i + \sum_{i=b+1}^{|\mathcal{S}|} \lambda_i \alpha_i - (k - h)\mu.$$

Using $\sum_{i=1}^{|\mathcal{S}|} \alpha_i = k$, we can rewrite

$$k - h = \sum_{i=1}^{|\mathcal{S}|} \alpha_i - h = \sum_{i=1}^{h}(\alpha_i - 1) + \sum_{i=h+1}^{b} \alpha_i + \sum_{i=b+1}^{|\mathcal{S}|} \alpha_i.$$

Substituting this identity:

$$\operatorname{tr}(A\Pi_{\widehat{E}}) - \sum_{i=1}^{k} \lambda_i = \sum_{i=1}^{h} \lambda_i(\alpha_i - 1) + \sum_{i=h+1}^{b} \mu\alpha_i + \sum_{i=b+1}^{|\mathcal{S}|} \lambda_i\alpha_i$$

$$- \mu\left[\sum_{i=1}^{h}(\alpha_i - 1) + \sum_{i=h+1}^{b} \alpha_i + \sum_{i=b+1}^{|\mathcal{S}|} \alpha_i\right]$$

$$= \sum_{i=1}^{h}(\mu - \lambda_i)(1 - \alpha_i) + \sum_{i=b+1}^{|\mathcal{S}|} (\lambda_i - \mu)\alpha_i.$$

Returning to the definition of $\alpha_i$, this is

$$\operatorname{tr}(A\Pi_{\widehat{E}}) - \sum_{i=1}^{k} \lambda_i = \sum_{i=1}^{h}(\mu - \lambda_i)\big(1 - \langle e_i, \Pi_{\widehat{E}}e_i\rangle\big) + \sum_{i=b+1}^{|\mathcal{S}|} (\lambda_i - \mu)\langle e_i, \Pi_{\widehat{E}}e_i\rangle.$$

The second term is nonnegative, and $\mu - \lambda_i \geq \mu - \lambda_h$ for $i \leq h$. Hence

$$(\mu - \lambda_h) \sum_{i=1}^{h}\big(1 - \langle e_i, \Pi_{\widehat{E}}e_i\rangle\big) < \epsilon.$$

Equivalently, since $(e_i)_{i=1}^{|\mathcal{S}|}$ is an orthonormal basis and $\Pi_E$ is the orthogonal projector onto $E = \operatorname{span}(e_1, \ldots, e_h)$,

$$\|(I - \Pi_{\widehat{E}})\Pi_E\|_{\mathrm{F}}^2 = \sum_{i=1}^{|\mathcal{S}|} \|(I - \Pi_{\widehat{E}})\Pi_E e_i\|_2^2$$

$$= \sum_{i=1}^{h} \|(I - \Pi_{\widehat{E}})e_i\|_2^2 + \sum_{i=h+1}^{|\mathcal{S}|} \|(I - \Pi_{\widehat{E}})0\|_2^2$$

$$= \sum_{i=1}^{h} \|(I - \Pi_{\widehat{E}})e_i\|_2^2.$$

Moreover, since $\Pi_{\widehat{E}}$ is an orthogonal projector, $I - \Pi_{\widehat{E}}$ is also an orthogonal projector. Hence, for every $i \leq h$,

$$\|(I - \Pi_{\widehat{E}})e_i\|_2^2 = \langle(I - \Pi_{\widehat{E}})e_i, (I - \Pi_{\widehat{E}})e_i\rangle$$

$$= \langle e_i, (I - \Pi_{\widehat{E}})e_i\rangle$$

$$= 1 - \langle e_i, \Pi_{\widehat{E}}e_i\rangle.$$

Therefore

$$\|(I - \Pi_{\widehat{E}})\Pi_E\|_{\mathrm{F}}^2 = \sum_{i=1}^{h}\big(1 - \langle e_i, \Pi_{\widehat{E}}e_i\rangle\big)$$

$$< \frac{\epsilon}{\mu - \lambda_h} = \frac{\epsilon}{\lambda_{h+1} - \lambda_h}.$$

We now construct an $h$-dimensional comparison subspace inside $\widehat{E}$. Recall that $E = \operatorname{span}(e_1, \ldots, e_h)$ and $\widehat{E} = \operatorname{span}(\widehat{e}_1, \ldots, \widehat{e}_k)$, with $h \leq k$. From the previous estimate we have

$$\|(I - \Pi_{\widehat{E}})\Pi_E\|_{\mathrm{F}}^2 < \frac{\epsilon}{\lambda_{h+1} - \lambda_h}.$$

We first consider the case

$$\frac{\epsilon}{\lambda_{h+1} - \lambda_h} < 1.$$

Then

$$\|(I - \Pi_{\widehat{E}})\Pi_E\|_2 \leq \|(I - \Pi_{\widehat{E}})\Pi_E\|_{\mathrm{F}} < 1.$$

We claim that the restriction $\Pi_{\widehat{E}}|_E : E \to \widehat{E}$ is injective. Indeed, let $x \in E$ and suppose that $\Pi_{\widehat{E}} x = 0$. Since $x \in E$, we have $\Pi_E x = x$. Therefore

$$x = (I - \Pi_{\widehat{E}})x = (I - \Pi_{\widehat{E}})\Pi_E x.$$

Taking norms gives

$$\|x\|_2 = \|(I - \Pi_{\widehat{E}})\Pi_E x\|_2 \leq \|(I - \Pi_{\widehat{E}})\Pi_E\|_2 \|x\|_2 < \|x\|_2.$$

This is impossible unless $x = 0$. Hence the kernel of $\Pi_{\widehat{E}}|_E$ is trivial, so $\Pi_{\widehat{E}}|_E$ is injective. Since $E$ is finite-dimensional and $\dim(E) = h$, it follows that

$$\dim(\Pi_{\widehat{E}} E) = \dim(E) = h.$$

In this case we define

$$\widehat{E}_h := \Pi_{\widehat{E}} E.$$

Then $\widehat{E}_h \subseteq \widehat{E}$ and $\dim(\widehat{E}_h) = h$.

We now bound the distance between $E$ and $\widehat{E}_h$. For every $x \in E$, the vector $\Pi_{\widehat{E}} x$ belongs to $\widehat{E}_h$. Hence the best approximation of $x$ from $\widehat{E}_h$ cannot be worse than the particular approximation $\Pi_{\widehat{E}} x$. Therefore,

$$\|(I - \Pi_{\widehat{E}_h})x\|_2 = \inf_{y \in \widehat{E}_h} \|x - y\|_2 \leq \|x - \Pi_{\widehat{E}} x\|_2 = \|(I - \Pi_{\widehat{E}})x\|_2.$$

Since this holds for all $x \in E$, we obtain

$$\|(I - \Pi_{\widehat{E}_h})\Pi_E\|_2 \leq \|(I - \Pi_{\widehat{E}})\Pi_E\|_2 \leq \|(I - \Pi_{\widehat{E}})\Pi_E\|_{\mathrm{F}}.$$

Lemma C.1 gives

$$\|\Pi_E - \Pi_{\widehat{E}_h}\|_2 = \|(I - \Pi_{\widehat{E}_h})\Pi_E\|_2.$$

Combining this identity with the previous estimate yields

$$\|\Pi_E - \Pi_{\widehat{E}_h}\|_2 \leq \|(I - \Pi_{\widehat{E}})\Pi_E\|_2 \leq \|(I - \Pi_{\widehat{E}})\Pi_E\|_{\mathrm{F}}.$$

Consequently,

$$\|\Pi_E - \Pi_{\widehat{E}_h}\|_2 \leq \|(I - \Pi_{\widehat{E}})\Pi_E\|_{\mathrm{F}} \leq \sqrt{\frac{2\epsilon}{\lambda_{h+1} - \lambda_h}}.$$

It remains to consider the case

$$\frac{\epsilon}{\lambda_{h+1} - \lambda_h} \geq 1.$$

Since $\dim(\widehat{E}) = k \geq h$, we may choose any $h$-dimensional subspace $\widehat{E}_h \subseteq \widehat{E}$. The matrices $\Pi_E$ and $\Pi_{\widehat{E}_h}$ are Euclidean orthogonal projectors onto subspaces of the same dimension. Hence

$$\|\Pi_E - \Pi_{\widehat{E}_h}\|_2 \leq 1.$$

Since $\epsilon/(\lambda_{h+1} - \lambda_h) \geq 1$, we have

$$1 < \sqrt{\frac{2\epsilon}{\lambda_{h+1} - \lambda_h}},$$

and therefore

$$\|\Pi_E - \Pi_{\widehat{E}_h}\|_2 < \sqrt{\frac{2\epsilon}{\lambda_{h+1} - \lambda_h}}.$$

Thus, in all cases, there exists an $h$-dimensional subspace $\widehat{E}_h \subseteq \widehat{E}$ such that

$$\|\Pi_E - \Pi_{\widehat{E}_h}\|_2 < \sqrt{\frac{2\epsilon}{\lambda_{h+1} - \lambda_h}}.$$

Returning to the original $\Phi$-weighted coordinates, define

$$\widehat{H} := \Phi^{-1/2}\widehat{E}_h.$$

Then $\widehat{H}$ is an $h$-dimensional subspace of $\widehat{S} = \operatorname{span}(\widehat{u}_1, \dots, \widehat{u}_k)$, and its $\Phi$-orthogonal projector satisfies

$$\|\Pi_H^\Phi - \Pi_{\widehat{H}}^\Phi\|_\Phi < \sqrt{\frac{2\epsilon}{\lambda_{h+1} - \lambda_h}}.$$

Since $\widehat{H} \subseteq \widehat{S}$, the least-squares error over $\widehat{S}$ is no larger than the least-squares error over $\widehat{H}$:

$$\|v - \widehat{v}_k\|_\Phi = \|v - \Pi_{\widehat{S}}^\Phi v\|_\Phi \leq \|v - \Pi_{\widehat{H}}^\Phi v\|_\Phi.$$

Hence

$$\begin{aligned}
\|v - \widehat{v}_k\|_\Phi &\leq \|v - \Pi_{\widehat{H}}^\Phi v\|_\Phi \\
&\leq \|v - \Pi_H^\Phi v\|_\Phi + \|(\Pi_H^\Phi - \Pi_{\widehat{H}}^\Phi)v\|_\Phi \\
&\leq \|v - \Pi_H^\Phi v\|_\Phi + \|\Pi_H^\Phi - \Pi_{\widehat{H}}^\Phi\|_\Phi\|v\|_\Phi.
\end{aligned}$$

The first term is the truncation error obtained by projecting onto $\operatorname{span}(u_1, \dots, u_h)$. By Lemma 3.4,

$$\|v - \Pi_H^\Phi v\|_\Phi \leq \|\bar{r}\|_\Phi \sqrt{\frac{1}{\lambda_2 \lambda_{h+1}}}.$$

Combining the two estimates gives

$$\|v - \widehat{v}_k\|_\Phi \leq \|\bar{r}\|_\Phi \sqrt{\frac{1}{\lambda_2 \lambda_{h+1}}} + \|v\|_\Phi \sqrt{\frac{2\epsilon}{\lambda_{h+1} - \lambda_h}},$$

as claimed. $\qquad\square$

**Lemma 3.4.** *Let $v_k$ be the approximation of $v$ produced by the $\Phi$-weighted linear least-squares oracle (Assumption 3.2) using the (exact) $k$ eigenvectors of $L$ associated with the $k$ smallest eigenvalues. Then:*

$$\|v - v_k\|_\Phi^2 \leq \frac{\|\bar{r}\|_\Phi^2}{\lambda_2 \lambda_{k+1}}$$

*where $0 = \lambda_1 \leq \lambda_2 \leq \dots \leq \lambda_{|\mathcal{S}|}$.*

*Proof.* First we write $L$ as:

$$L = I - \frac{P + \Phi^{-1}P^\top \Phi}{2} := I - \frac{P + P^*}{2} = I - R.$$

$R$ is $\Phi$-self-adjoint so $\Phi R = R^\top \Phi$, since:

$$R^* := \left(\frac{P + P^*}{2}\right)^* = \left(\frac{P^* + (P^*)^*}{2}\right) = R.$$

Let $\mathcal{L} = \Phi^{\frac{1}{2}}L\Phi^{-\frac{1}{2}} = I - \Phi^{\frac{1}{2}}R\Phi^{-\frac{1}{2}} = I - S$. $\mathcal{L}$ is semi-definite positive for similarity and also symmetric:

$$S^\top = (\Phi^{\frac{1}{2}}R\Phi^{-\frac{1}{2}})^\top = \Phi^{-\frac{1}{2}}R^\top \Phi^{\frac{1}{2}} = \Phi^{-\frac{1}{2}}(\Phi R\Phi^{-1})\Phi^{\frac{1}{2}} = \Phi^{\frac{1}{2}}R\Phi^{-\frac{1}{2}}.$$

Define $x := \Phi^{\frac{1}{2}} v$. Let $\Psi = (u_1, \ldots, u_k)$ where $u_i$ are eigenvectors of $L$ associated with the $k$ lowest eigenvalues, and $G := \Psi^\top \Phi \Psi$. Define the $\Phi$-orthogonal projection onto $\text{span}(\Psi)$ by

$$v_k := \Psi(\Psi^\top \Phi \Psi)^{-1} \Psi^\top \Phi\, v = \Psi G^{-1} \Psi^\top \Phi v, \qquad x_k := \Phi^{\frac{1}{2}} v_k.$$

Set $Q := \Phi^{1/2} \Psi$. Then $Q$'s columns are eigenvectors of $\mathcal{L}$ (since $Lu_i = \lambda_i u_i \Rightarrow \mathcal{L}(\Phi^{1/2} u_i) = \lambda_i (\Phi^{1/2} u_i)$). Moreover

$$x_k = \Phi^{1/2} \Psi G^{-1} \Psi^\top \Phi v = (\Phi^{1/2}\Psi)(\Psi^\top \Phi \Psi)^{-1}(\Psi^\top \Phi^{1/2})(\Phi^{1/2} v) = Q(Q^\top Q)^{-1} Q^\top x.$$

Hence $x_k$ is the (Euclidean) orthogonal projection of $x$ onto $\text{span}(Q) = \text{span}(\Phi^{1/2} u_1, \ldots, \Phi^{1/2} u_k)$, i.e. the subspace of $\mathcal{L}$ associated with the $k$ lowest eigenvalues. Therefore, by Lemma A.4,

$$\|x - x_k\|_2^2 \leq \frac{x^T \mathcal{L} x}{\lambda_{k+1}(\mathcal{L})} = \frac{x^T \mathcal{L} x}{\lambda_{k+1}(L)}.$$

Now, consider the vector space $\mathcal{V} = \{v \in \mathbb{R}^{|\mathcal{S}|} : \phi^\top v = 0\}$ equipped with the $\Phi$-scalar product: $\langle u, v \rangle_\Phi = u^T \Phi v$, $\forall u, v \in \mathcal{V}$:

$$\|v - v_k\|_\Phi^2 = \|\Phi^{\frac{1}{2}} v - \Phi^{\frac{1}{2}} v_k\|_2^2 = \|x - x_k\|_2^2.$$

So we have that:

$$\|v - v_k\|_\Phi^2 \leq \frac{x^T \mathcal{L} x}{\lambda_{k+1}(L)} = \frac{v^\top \Phi(I - R)v}{\lambda_{k+1}(L)} = \frac{v^\top \Phi L v}{\lambda_{k+1}(L)}.$$

In this space the inverse of $(I - P)$ and $(I - R)$ are well defined since they are bijective when restricted to this space (Lemma A.5). We omit the notation $|_\mathcal{V}$ to indicate the restriction to $\mathcal{V}$ but it's implicit throughout the proof on all the operators.

$$
\begin{aligned}
v^T \Phi(I - R)v &= \langle v, (I - R)v \rangle_\Phi \\
&= \frac{1}{2} \langle v, (2I - P - P^*)v \rangle_\Phi \\
&= \frac{1}{2}\big[\langle v, (I - P)v \rangle_\Phi + \langle v, (I - P^*)v \rangle_\Phi\big].
\end{aligned}
$$

But actually $\langle v, (I - P)v \rangle_\Phi$ and $\langle v, (I - P^*)v \rangle_\Phi$ are the same quantity. More in general:

$$\langle v, A^* v \rangle_\Phi = v^T \Phi A^* v = v^T \Phi \Phi^{-1} A^T \Phi v = v^T A^T \Phi v = \langle Av, v \rangle_\Phi = \langle v, Av \rangle_\Phi. \tag{13}$$

So we have that:

$$v^T \Phi(I - R)v = \langle v, (I - P)v \rangle_\Phi = \langle v, \bar{r} \rangle_\Phi.$$

Now in the space $\mathcal{V}$ we can write:

$$v^T \Phi(I - R)v = \langle v, \bar{r} \rangle_\Phi = \langle (I - P)^{-1} \bar{r}, \bar{r} \rangle_\Phi.$$

Let $\text{sym}_\Phi(A) = \frac{A + A^*}{2}$, using (13) we have that:

$$\langle (I - P)^{-1} \bar{r}, \bar{r} \rangle_\Phi = \langle \bar{r}, (I - P)^{-1} \bar{r} \rangle_\Phi = \langle \bar{r}, \text{sym}_\Phi\big[(I - P)^{-1}\big] \bar{r} \rangle_\Phi.$$

In this way:

$$\langle \bar{r}, \text{sym}_\Phi\big[(I - P)^{-1}\big] \bar{r} \rangle_\Phi \leq \langle \bar{r}, \big[\text{sym}_\Phi(I - P)\big]^{-1} \bar{r} \rangle_\Phi = \langle \bar{r}, (I - R)^{-1} \bar{r} \rangle_\Phi = \bar{r}^\top \Phi L^{-1} \bar{r}.$$

First observe that

$$\phi^\top \bar{r} = \phi^\top \big(r - \mathbf{1}\, r^\top \phi\big) = \phi^\top r - (\phi^\top \mathbf{1})\, r^\top \phi = \phi^\top r - 1 \cdot \phi^\top r = 0,$$

hence $\bar{r} \in \mathcal{V}$. By Lemma A.5, $L$ is bijective on $\mathcal{V}$, so $L^{-1}\bar{r}$ is well-defined.

Moreover,

$$(\Phi^{1/2}\mathbf{1})^\top(\Phi^{1/2}\bar{r}) = \mathbf{1}^\top\Phi\bar{r} = \phi^\top\bar{r} = 0,$$

so $\Phi^{1/2}\bar{r} \in \ker(\mathcal{L})^\perp$ and $\mathcal{L}^{-1}$ is well-defined on $\Phi^{1/2}\bar{r}$. Using $L^{-1} = \Phi^{-1/2}\mathcal{L}^{-1}\Phi^{1/2}$ on $\mathcal{V}$,

$$\bar{r}^\top\Phi L^{-1}\bar{r} = \bar{r}^\top\Phi\Phi^{-1/2}\mathcal{L}^{-1}\Phi^{1/2}\bar{r} = (\Phi^{1/2}\bar{r})^\top\mathcal{L}^{-1}(\Phi^{1/2}\bar{r}) \leq \frac{\|\Phi^{1/2}\bar{r}\|_2^2}{\lambda_2(\mathcal{L})} = \frac{\|\bar{r}\|_\Phi^2}{\lambda_2(L)}.$$

Therefore,

$$v^\top\Phi Lv = v^\top\Phi(I-R)v \leq \frac{\|\bar{r}\|_\Phi^2}{\lambda_2(L)}.$$

Combining with the previous estimate yields

$$\|v - v_k\|_\Phi^2 \leq \frac{v^\top\Phi Lv}{\lambda_{k+1}(L)} \leq \frac{\|\bar{r}\|_\Phi^2}{\lambda_2(L)\lambda_{k+1}(L)}.$$

$\square$

**Lemma A.2.** $\langle v, sym_\Phi[(I-P)|_\mathcal{V}^{-1}]v\rangle \leq \langle v, [sym_\Phi(I-P)]|_\mathcal{V}^{-1}v\rangle, \quad \forall v \in \mathcal{V}$

*Proof.* The idea is to use Lemma A.3, which works in a Euclidean space, in our setting where we use $\Phi$-induced norm $\langle\cdot,\cdot\rangle_\Phi$.
To do this we define:

$$T := \Phi^{\frac{1}{2}}|_\mathcal{V} : \mathcal{V} \to \mathcal{W}$$

T is an isometry between $\langle\mathcal{V},\langle\cdot,\cdot\rangle_\Phi\rangle$ and $\langle\mathcal{W},\langle\cdot,\cdot\rangle_2\rangle$:

$$\langle x,y\rangle_\Phi = x^\top\Phi^{\frac{1}{2}}\Phi^{\frac{1}{2}}y = \langle Tx, Ty\rangle_2$$

Define $B$:

$$B := T(I-P)|_\mathcal{V}T^{-1} : \mathcal{W} \to \mathcal{W}$$

To use Lemma A.3 on $B$, we need $B$ to be invertible (Lemma A.5), and $sym(B) \succ 0$ on $\mathcal{W}$. To see the latter, we show that $sym(B)$ is similar to $sym(I-P)_\Phi|_\mathcal{V}$:

$$\begin{aligned}
T^{-1}sym(B)T &= T^{-1}\left(\frac{B+B^\top}{2}\right)T \\
&= T^{-1}\left(\frac{T(I-P)|_\mathcal{V}T^{-1} + T^{-1}(I-P)|_\mathcal{V}^\top T}{2}\right)T \\
&= sym_\Phi\big[(I-P)|_\mathcal{V}\big] = \big[sym_\Phi(I-P)\big]\big|_\mathcal{V}
\end{aligned} \tag{14}$$

and for Lemma A.6, $sym(B) \succ 0$ on $\mathcal{W}$. So for $sym(B)$, Lemma A.3 holds:

$$\langle y, sym(B^{-1})y\rangle \leq \langle y, sym(B)^{-1}y\rangle, \quad \forall y \in \mathcal{W}$$

Analogously to (14) we also have:

$$T^{-1}sym(B^{-1})T = sym_\Phi\big[(I-P)^{-1}\big|_\mathcal{V}\big]$$

Now we want to prove that:

$$\langle v, sym_\Phi\big[(I-P)^{-1}|_\mathcal{V}\big]v\rangle_\Phi = \langle y, sym(B^{-1})y\rangle$$

and:

$$\langle v, \big[sym_\Phi(I-P)^{-1}\big]\big|_\mathcal{V}v\rangle_\Phi = \langle y, sym(B)^{-1}y\rangle$$

which implies the thesis. Indeed:

$$\begin{aligned}
\langle v, sym_\Phi\big[(I-P)^{-1}|_\mathcal{V}\big]v\rangle_\Phi &= v^\top T^2 T^{-1}sym(B^{-1})Tv \\
&= \langle Tv, sym(B^{-1})Tv\rangle \\
&= \langle y, sym(B^{-1})y\rangle
\end{aligned}$$

and from (14), taking the inverse:

$$\langle v, \left[\text{sym}_\Phi(I-P)\right]^{-1}\big|_{\mathcal{V}} v \rangle_\Phi = v^\top T^2 \left[\text{sym}_\Phi(I-P)\right]^{-1}\big|_{\mathcal{V}} v$$
$$= v^\top T^2 T^{-1} \text{sym}(B)^{-1} T v$$
$$= \langle Tv, \text{sym}(B)^{-1} Tv \rangle$$
$$= \langle y, \text{sym}(B)^{-1} y \rangle$$

$\square$

**Lemma A.3.** *Let $A \in \mathbb{R}^{S \times S}$ be an invertible matrix such that $sym(A) = \frac{A+A^\top}{2} \succ 0$, then:*

$$sym(A^{-1}) \preceq \left[sym(A)\right]^{-1}$$

*Proof.* See Yang & Lu (2018), Lemma 2 or Johnson (1973), Theorem 2 by setting c = 0. $\square$

**Lemma A.4.** *Let $v \in \mathbb{R}^{|\mathcal{S}|}$, $\mathcal{L}$ symmetric and positive definite matrix, $v_k$ the projection of $v$ onto the $k$ eigenvectors associated with the $k$-lowest eigenvalues of $\mathcal{L}$. Then:*

$$\|v - v_k\|_2^2 \leq \frac{v^\top \mathcal{L} v - v_k^\top \mathcal{L} v_k}{\lambda_{k+1}(\mathcal{L})}$$

*Proof.* The eigenvectors of $\mathcal{L}$, $u_1, \ldots, u_{|\mathcal{S}|}$ are such that $\text{span}(u_1, \ldots, u_{|\mathcal{S}|}) = \mathcal{V}$ and since ($v \in \mathcal{V}$, we have that $v$ can be expressed as linear combination of $u_1, \ldots, u_{|\mathcal{S}|}$:

$$v = \sum_{i=1}^{|\mathcal{S}|} a_i u_i, \qquad v_k = \sum_{i=1}^{k} a_i u_i$$

This leads to:

$$\|v - v_k\|_2^2 = \sum_{i=k+1}^{|\mathcal{S}|} a_i^2$$

The for what concerns the square of the $L$-induced norm of $v$, $v^\top L v$:

$$v^\top L v = v^\top \sum_{i=1}^{|\mathcal{S}|} \lambda_i u_i u_i^\top v = \sum_{i=1}^{|\mathcal{S}|} \lambda_i a_i^2$$

To conclude we have that:

$$\|v - v_k\|_2^2 = \sum_{i=k+1}^{|\mathcal{S}|} a_i^2 \leq \frac{\sum_{i=k+1}^{|\mathcal{S}|} \lambda_i a_i^2}{\lambda_{k+1}(\mathcal{L})} \leq \frac{v^\top \mathcal{L} v - v_k^\top \mathcal{L} v_k}{\lambda_{k+1}(\mathcal{L})}$$

$\square$

**Lemma A.5.** *Given $P \in \mathbb{R}^{S \times S}$ stochastic ergodic with stationary distribution $\phi$, let $R = \frac{(P+P^*)}{2}$ where $P^* = \Phi^{-1} P^\top \Phi$ and $\Phi = diag(\phi)$. Then $I - P$ and $I - R$ are bijective linear maps when restricted to $\mathcal{V} = \{v \in \mathbb{R}^{|\mathcal{S}|} | \phi^\top v = 0\}$.*

*Proof.* We first need to prove that $\mathcal{V}$ is invariant under $I - P$ and $I - R$, namely $\forall v \in \mathcal{V}$, $(I-P)v \in \mathcal{V}$ and $(I-R)v \in \mathcal{V}$.

$$\phi^\top(I-P)v = \phi^\top v - \phi^\top P v = \phi^\top v - \phi^\top v = 0, \quad \forall v \in \mathcal{V}$$

$$\phi^\top(I-R)v = \frac{1}{2}\phi^\top(I-P)v + \frac{1}{2}\phi^\top(I-P^*)v$$
$$= 0 + \frac{1}{2}\phi^\top(I - \Phi^{-1}P^\top\Phi)v = \frac{1}{2}(\phi^\top v - \phi^T v) = 0, \quad \forall v \in \mathcal{V}$$

The last inequality is due to:

$$\phi^\top(\Phi^{-1}P^\top\Phi) = x^\top = \phi^T$$

since:

$$x_j^\top = \sum_{i=1}^{|\mathcal{S}|} \phi_i \frac{1}{\phi_i}(P^\top)_{ij}\phi_j = \sum_{i=1}^{|\mathcal{S}|} P_{ji}\phi_j = \phi_j^\top$$

Now to prove the invertibility of $I - P$ and $I - R$ in $\mathcal{V}$ it suffices to prove they are injective.

Let's start with $I - P$, which we denote as $(I - P)|_{\mathcal{V}} : \mathcal{V} \to \mathcal{V}$ when restricted to $\mathcal{V}$. The injectivity of $(I - P)|_{\mathcal{V}}$ is equivalent to $\text{Ker}((I - P)|_{\mathcal{V}}) = \{\mathbf{0}\}$.

Let:

$$(I - P)v = 0, \quad v \in \mathcal{V}$$

So $v$ is an eigenvector of $P$ associated with the Perron eigenvalue $\lambda_{\mathcal{S}} = 1$ and, given that $P$ is ergodic, $\lambda_{\mathcal{S}}$ is simple and its eigenspace is $\text{span}(\mathbf{1})$. This means that $v = c\mathbf{1}$ for some $c \in \mathbb{R}$.

Now, since $v \in \mathcal{V}$:

$$0 = \phi^\top v = \phi^\top \mathbf{1} c = c$$

so $v = \mathbf{0}$, thus $\text{Ker}((I - P)|_{\mathcal{V}}) = \{\mathbf{0}\}$. Now for $(I - R)|_{\mathcal{V}}$, it's analogous since $R$ is ergodic as well. $\qquad\square$

**Lemma A.6.** $\left[sym_\Phi(I - P)\right]\big|_{\mathcal{V}} \succ 0$ in $\langle \mathcal{V}, \langle \cdot, \cdot \rangle_\Phi \rangle$

*Proof.* We need to prove that:

$$\langle v, \text{sym}_\Phi(I - P)v \rangle_\Phi > 0, \quad \forall v \in \mathcal{V}$$

We know that:

$$\langle v, \text{sym}_\Phi(I - P)v \rangle_\Phi \geq 0, \quad \forall v \in \mathbb{R}^{|\mathcal{S}|} \tag{15}$$

since:

$$\langle v, \text{sym}_\Phi(I - P)v \rangle_\Phi = \langle v, \Phi(I - R)v \rangle = \langle v, Lv \rangle$$

and $L \succeq 0$. Also $\ker(\text{sym}_\Phi(I - P)) = \text{span}(\mathbf{1})$.

Suppose by contradiction that $v \in \mathcal{V}$ with $v \neq \mathbf{0}$ is such that:

$$\langle v, \text{sym}_\Phi(I - P)v \rangle_\Phi = 0$$

This means that:

$$\|\text{sym}_\Phi(I - P)^{\frac{1}{2}}v\|_\Phi = 0 \iff \text{sym}_\Phi(I - P)v = \mathbf{0}$$

thus:

$$v \in \text{span}(\mathbf{1})$$

but $v \in \mathcal{V}$, so $\phi^\top v = 0$, which means that:

$$\phi^\top c\mathbf{1} = 0$$

for some $c \in \mathbb{R}$. Now given that $\phi^\top \mathbf{1} = 1$, we conclude that $c = 0$, and so $v = \mathbf{0}$ which contradicts the hypothesis, thus the thesis.

$\qquad\square$

**Proposition A.7.** *Let $\phi$ be the the stationary distribution of the Markov chain induced by $\mathcal{P}^\pi$ and define $D(u, v)$:*

$$D(u, v) = \frac{1}{2}\frac{d\phi(u)}{d\mu(u)}\frac{dP(v \mid u)}{d\mu(v)} + \frac{1}{2}\frac{d\phi(v)}{d\mu(v)}\frac{dP(u \mid v)}{d\mu(u)}$$

*Let $L$ be the linear, self-adjoint (Hilbert-Schmidt integral operator thus compact) operator:*

$$Lf(u) = f(u)\int_S D(u, v)d\mu(v) - \int_S f(v)D(u, v)\,d\mu(v).$$

*Then:*

$$\sum_{i=1}^k \langle f_i, Lf_i \rangle_{\mathcal{H}} = \frac{1}{2}\int_S \int_S \sum_{i=1}^k (f_i(u) - f_i(v))^2 D(u, v)\,d\mu(u)\,d\mu(v) = \mathbb{E}_{u \sim \phi, v \sim \mathcal{P}^\pi(\cdot|u)}\left[\sum_{i=1}^k \left(f_i(u) - f_i(v)\right)^2\right]$$

*Proof.* For the first equality:

$$\langle f, Lf \rangle_{\mathcal{H}} = \int_S f(u) \left( f(u) \int_S D(u,v) d\mu(v) - \int_S f(v) D(u,v) d\mu(v) \right) d\mu(u) =$$

$$= \int_S f(u) \int_S \left( f(u) - f(v) \right) D(u,v) d\mu(v) d\mu(u) =$$

$$= \int_S \int_S f(u)^2 D(u,v) d\mu(u) d\mu(v) - \int_S \int_S f(u) f(v) D(u,v) d\mu(u) d\mu(v) =$$

$$= \frac{1}{2} \int_S \int_S f(u)^2 D(u,v) d\mu(v) d\mu(u) + \frac{1}{2} \int_S \int_S f(v)^2 D(v,u) d\mu(u) d\mu(v) -$$

$$- \int_S \int_S f(u) f(v) D(u,v) d\mu(u) d\mu(v) =$$

$$= \frac{1}{2} \int_S \int_S \left( f(u) - f(v) \right)^2 D(u,v) \, d\mu(u) \, d\mu(v)$$

So for $\sum_{i=1}^k \langle f_i, Lf_i \rangle_{\mathcal{H}}$:

$$\sum_{i=1}^k \langle f_i, Lf_i \rangle_{\mathcal{H}} = \frac{1}{2} \int_S \int_S \sum_{i=1}^k (f_i(u) - f_i(v))^2 D(u,v) \, d\mu(u) \, d\mu(v)$$

Then by defining:

$$D(u,v) = \frac{1}{2} \frac{d\phi(u)}{d\mu(u)} \frac{dP(v \mid u)}{d\mu(v)} + \frac{1}{2} \frac{d\phi(v)}{d\mu(v)} \frac{dP(u \mid v)}{d\mu(u)},$$

we have that:

$$\sum_{i=1}^k \langle f_i, Lf_i \rangle_{\mathcal{H}} = \frac{1}{2} \int_S \int_S \sum_{i=1}^k (f_i(u) - f_i(v))^2 D(u,v) \, d\mu(u) \, d\mu(v) =$$

$$= \frac{1}{4} \int_S \int_S \sum_{i=1}^k \left( f_i(u) - f_i(v) \right)^2 \frac{d\phi(u)}{d\mu(u)} \frac{dP(v \mid u)}{d\mu(v)} d\mu(u) \, d\mu(v) +$$

$$+ \frac{1}{4} \int_S \int_S \sum_{i=1}^k \left( f_i(u) - f_i(v) \right)^2 \frac{d\phi(v)}{d\mu(v)} \frac{dP(u \mid v)}{d\mu(u)} d\mu(u) \, d\mu(v) =$$

$$= \frac{1}{2} \int_S \int_S \sum_{i=1}^k \left( f_i(u) - f_i(v) \right)^2 dP(v \mid u) d\phi(u) = \mathbb{E}_{u \sim \phi, v \sim P(\cdot \mid u)} \left[ \sum_{i=1}^k \left( f_i(u) - f_i(v) \right)^2 \right]$$

$\square$

**Lemma 3.5.** *[Graph Drawing Lemma] Let $u_1, \ldots, u_{|\mathcal{S}|}$ be the eigenvectors of a p.s.d. symmetric matrix $A$ associated with eigenvalues $0 = \lambda_1(A) < \lambda_2(A) \leq \cdots \leq \lambda_{|\mathcal{S}|}(A)$. Let $\tilde{u}_1, \ldots, \tilde{u}_k \in \mathbb{R}^{|\mathcal{S}|}$ be orthonormal vectors satisfying $\tilde{u}_i^\top \tilde{u}_j = \delta_{ij}$ for all $i, j \in \{1, \ldots, k\}$.*

*Define $\Psi := (u_1, \ldots, u_k) \in \mathbb{R}^{|\mathcal{S}| \times k}$ and $\tilde{\Psi} := (\tilde{u}_1, \ldots, \tilde{u}_k) \in \mathbb{R}^{|\mathcal{S}| \times k}$, and let $\Pi_\Psi$ and $\Pi_{\tilde{\Psi}}$ denote the orthogonal projection matrices onto $\text{span}(\Psi)$ and $\text{span}(\tilde{\Psi})$, respectively.*

*Suppose there exists $\epsilon > 0$ such that*

$$\sum_{i=1}^k \tilde{u}_i^\top A \tilde{u}_i - \sum_{i=1}^k \lambda_i \; < \; \epsilon.$$

*Then,*

$$\|\Pi_\Psi - \Pi_{\tilde{\Psi}}\|_2 \; < \; \sqrt{\frac{2\epsilon}{\lambda_{k+1}(A) - \lambda_k(A)}}.$$

*Proof.* Let $\mathcal{U} = \mathrm{span}(\Psi)$ and $\tilde{\mathcal{U}} = \mathrm{span}(\tilde{\Psi})$. Since $A$ is symmetric, we have that its eigenvectors span $\mathbb{R}^{|\mathcal{S}|}$, thus:

$$\mathbb{R}^{|\mathcal{S}|} = \mathcal{U} \oplus \mathcal{U}^{\perp}.$$

Each $\tilde{u}_i$ can be decomposed into the sum of a component in $\mathcal{U}$ and another in $\mathcal{U}^{\perp}$:

$$\tilde{u}_i = \tilde{u}_i^{\|} + \tilde{u}_i^{\perp}$$

where $\tilde{u}_i^{\|} \in \mathcal{U}$ and $\tilde{u}_i^{\perp} \in \mathcal{U}^{\perp}$ for all $i = 1, \dots, k$. In this way, the hypothesis can be rewritten as follows:

$$\sum_{i=1}^{k} \tilde{u}_i^{\|\top} A \tilde{u}_i^{\|} + \sum_{i=1}^{k} \tilde{u}_i^{\perp\top} A \tilde{u}_i^{\perp} - \sum_{i=1}^{k} \lambda_i < \epsilon.$$

We will first show that:

$$\sum_{i=1}^{k} \tilde{u}_i^{\|\top} A \tilde{u}_i^{\|} + \sum_{i=1}^{k} \tilde{u}_i^{\perp\top} A \tilde{u}_i^{\perp} - \sum_{i=1}^{k} \lambda_i \geq \sum_{i=1}^{k} (\lambda_{k+1} - \lambda_k) \|\tilde{u}_i^{\perp}\|^2, \tag{16}$$

so that:

$$\sum_{i=1}^{k} \|\tilde{u}_i^{\perp}\|^2 < \frac{\epsilon}{\lambda_{k+1} - \lambda_k}. \tag{17}$$

Then we will show that:

$$\sum_{i=1}^{k} \|\tilde{u}_i^{\perp}\|^2 = \|\mathbf{Sin\Theta}(\mathcal{U}, \tilde{\mathcal{U}})\|_F^2 \geq \frac{1}{2} \|\Pi_{\Psi} - \Pi_{\tilde{\Psi}}\|_2^2, \tag{18}$$

where the last inequality is by Lemma (C.1) together with the Frobenius-spectral norm inequality. Together, Equations (17) and (18) imply the thesis. So, we divide the rest of the proof in two steps.

**Step 1.** Let's prove Equation (16) first. Notice that:

$$\sum_{i=1}^{k} \tilde{u}_i^{\perp\top} A \tilde{u}_i^{\perp} \geq \sum_{i=1}^{k} \lambda_{k+1} \|\tilde{u}_i^{\perp}\|^2, \tag{19}$$

This holds since, for every $i$, every $\tilde{u}_i^{\perp}$ can be written as a linear combination of $u_{k+1}, \dots, u_{|\mathcal{S}|}$:

$$\tilde{u}_i^{\perp} = \sum_{m=k+1}^{|\mathcal{S}|} b_m u_m,$$

so that:

$$\begin{aligned}
\tilde{u}_i^{\perp\top} A \tilde{u}_i^{\perp} &= \left( \sum_{m=k+1}^{|\mathcal{S}|} b_m u_m \right)^{\top} A \left( \sum_{h=k+1}^{|\mathcal{S}|} b_h u_h \right) \\
&= \left( \sum_{m=k+1}^{|\mathcal{S}|} b_m u_m \right)^{\top} \left( \sum_{h=k+1}^{|\mathcal{S}|} \lambda_h b_h u_h \right) \\
&= \sum_{m=k+1}^{|\mathcal{S}|} \lambda_m b_m^2 \geq \lambda_{k+1} \sum_{m=k+1}^{|\mathcal{S}|} b_m^2 = \lambda_{k+1} \|\tilde{u}_i^{\perp}\|^2.
\end{aligned}$$

Given Equation (19), we just need to show that:

$$\sum_{i=1}^{k} \tilde{u}_i^{\|\top} A \tilde{u}_i^{\|} - \sum_{i=1}^{k} \lambda_i \geq \sum_{i=1}^{k} -\lambda_k \|\tilde{u}_i^{\perp}\|^2. \tag{20}$$

To prove the latter, we proceed by contradiction. Namely, suppose that:

$$\sum_{i=1}^{k} \tilde{u}_i^{\|\top} A \tilde{u}_i^{\|} < \sum_{i=1}^{k} (\lambda_i - \lambda_k \|\tilde{u}_i^{\perp}\|^2).$$

Since $\|\tilde{u}_i\|_2 = 1$ for all $i = 1, \ldots, k$, we have that:

$$\|\tilde{u}_i^{\perp}\|^2 = 1 - \|\tilde{u}_i^{\|}\|^2,$$

thus:

$$\sum_{i=1}^{k} \tilde{u}_i^{\|\top} A \tilde{u}_i^{\|} < \sum_{i=1}^{k} (\lambda_i - \lambda_k) + \sum_{i=1}^{k} \lambda_k \|\tilde{u}_i^{\|}\|^2. \tag{21}$$

Since $\tilde{u}_i^{\|} \in \mathcal{U}$, it can be written as a linear combination of $u_1, \ldots, u_k$:

$$\tilde{u}_i^{\|} = \alpha_{1i} u_1 + \alpha_{2i} u_2 + \ldots + \alpha_{ki} u_k,$$

which means:

$$\tilde{u}_i^{\|\top} A \tilde{u}_i^{\|} = (\alpha_{1i} u_1^{\top} + \ldots + \alpha_{ki} u_k^{\top})(\lambda_1 \alpha_{1i} u_1 + \ldots + \lambda_k \alpha_{ki} u_k) = \sum_{j=1}^{k} \lambda_j \alpha_{ji}^2.$$

Moreover:

$$\|\tilde{u}_i^{\|}\|_2^2 = \sum_{j=1}^{k} \alpha_{ji}^2.$$

By substituting the last two identities into Equation (21):

$$\sum_{i=1}^{k} \sum_{j=1}^{k} \lambda_j \alpha_{ji}^2 < \sum_{i=1}^{k} (\lambda_i - \lambda_k) + \sum_{i=1}^{k} \sum_{j=1}^{k} \lambda_k \alpha_{ji}^2.$$

Rearranging:

$$\sum_{i=1}^{k} \sum_{j=1}^{k} (\lambda_j - \lambda_k) \alpha_{ji}^2 < \sum_{i=1}^{k} (\lambda_i - \lambda_k),$$

$$\sum_{j=1}^{k} (\lambda_j - \lambda_k) \sum_{i=1}^{k} \alpha_{ji}^2 < \sum_{i=1}^{k} (\lambda_i - \lambda_k),$$

$$\sum_{j=1}^{k} (\lambda_k - \lambda_j) \sum_{i=1}^{k} \alpha_{ji}^2 > \sum_{j=1}^{k} (\lambda_k - \lambda_j). \tag{22}$$

Since $\lambda_k - \lambda_j \geq 0$ for all $j = 1, \ldots, k$, to show the contradiction we just need to prove that $\sum_{i=1}^{k} \alpha_{ji}^2 \leq 1$ for all $j = 1, \ldots, k$.

By defining $M \in \mathbb{R}^{k \times k}$ as the matrix containing the projections of $\tilde{u}_i$ on $\mathcal{U}$:

$$M = \begin{pmatrix} m_{11} & \cdots & m_{1k} \\ \vdots & \ddots & \vdots \\ m_{k1} & \cdots & m_{kk} \end{pmatrix} = \Psi^{\top} \begin{pmatrix} \tilde{u}_1 & \cdots & \tilde{u}_k \end{pmatrix} = \Psi^{\top} \tilde{\Psi}.$$

Denote by $M_j^{\top}$ the $j$-th row of $M$, so that $\|M_j^{\top}\|_2^2 = \sum_{i=1}^{k} m_{ji}^2$.

Note that $M_j^{\top} = e_j^{\top} M = e_j^{\top} \Psi^{\top} \tilde{\Psi}$, where $e_j$ is the $j$-th element of the canonical basis of $\mathbb{R}^k$. Hence:

$$\|M_j^{\top}\|_2^2 = e_j^{\top} \Psi^{\top} \tilde{\Psi} \tilde{\Psi}^{\top} \Psi e_j = \langle \Psi e_j, \Pi_{\tilde{\Psi}} \Psi e_j \rangle$$
$$\leq \|\Psi e_j\|_2 \|\Pi_{\tilde{\Psi}} \Psi e_j\|_2$$
$$\leq \|\Psi\|_2 \|e_j\|_2 \|\|\Pi_{\tilde{\Psi}}\|_2 \|\Psi\|_2 \|e_j\|_2 \leq 1$$

since $\Psi$ is orthogonal and $\Pi_{\tilde{\Psi}}$ is a projection matrix. This is in contradiction with Equation 22, proving that (20) holds, and consequently also (16) holds.

**Step 2.** To conclude the proof of the lemma, we just need to prove Equation (18). Notice that:

$$
\begin{aligned}
\sum_{i=1}^{k} \|\tilde{u}_i^\perp\|^2 &= \sum_{i=1}^{k} \|(I - \Pi_\Psi)\tilde{u}_i\|_2^2 \\
&= \sum_{i=1}^{k} \langle (I - \Pi_\Psi)\tilde{u}_i, (I - \Pi_\Psi)\tilde{u}_i \rangle \\
&= \sum_{i=1}^{k} \langle \tilde{u}_i, (I - \Pi_\Psi)\tilde{u}_i \rangle - \langle \Pi_\Psi \tilde{u}_i, (I - \Pi_\Psi)\tilde{u}_i \rangle \\
&= \sum_{i=1}^{k} \langle \tilde{u}_i, (I - \Pi_\Psi)\tilde{u}_i \rangle - \langle \tilde{u}_i^\|, \tilde{u}_i^\perp \rangle \\
&= \sum_{i=1}^{k} \langle \tilde{u}_i, (I - \Pi_\Psi)\tilde{u}_i \rangle.
\end{aligned}
$$

So we have that:

$$
\begin{aligned}
\sum_{i=1}^{k} \|\tilde{u}_i^\perp\|^2 &= \sum_{i=1}^{k} \tilde{u}_i^\top (I - \Pi_\Psi)\tilde{u}_i = \mathrm{Tr}\left( \sum_{i=1}^{k} \tilde{u}_i^\top (I - \Pi_\Psi)\tilde{u}_i \right) \\
&= \sum_{i=1}^{k} \mathrm{Tr}\left( \tilde{u}_i^\top (I - \Pi_\Psi)\tilde{u}_i \right) = \sum_{i=1}^{k} \mathrm{Tr}\left( \tilde{u}_i \tilde{u}_i^\top (I - \Pi_\Psi) \right) \\
&= \mathrm{Tr}\left( \sum_{i=1}^{k} \tilde{u}_i \tilde{u}_i^\top (I - \Pi_\Psi) \right) = \mathrm{Tr}\left( \Pi_{\tilde{\Psi}}(I - \Pi_\Psi) \right) \\
&= \mathrm{Tr}\left( \tilde{\Psi}\tilde{\Psi}^\top (I - \Psi\Psi^\top) \right) = \mathrm{Tr}(\tilde{\Psi}\tilde{\Psi}^\top) - \mathrm{Tr}(\tilde{\Psi}\tilde{\Psi}^\top \Psi\Psi^\top) \\
&= k - \mathrm{Tr}(\Psi^\top \tilde{\Psi}\tilde{\Psi}^\top \Psi) = k - \|\tilde{\Psi}^\top \Psi\|_F^2 \\
&= k - \sum_{m=1}^{k} \cos(\theta_m)^2 = \sum_{m=1}^{k} \sin(\theta_m)^2 = \|\mathbf{Sin\Theta}(\mathcal{U}, \tilde{\mathcal{U}})\|_F^2,
\end{aligned}
$$

which concludes the proof. $\qquad\square$

**Lemma 3.6.** *Let $u_1, \dots, u_{|\mathcal{S}|}$ be the eigenvectors of $L$ associated with eigenvalues $0 = \lambda_1 \leq \lambda_2 \leq \cdots \leq \lambda_{|\mathcal{S}|}$. Let $\widehat{u}_1, \dots, \widehat{u}_k \in \mathbb{R}^{|\mathcal{S}|}$ be their estimates produced by $\epsilon$-optimal GDO (Assumption 3.1). Define $\Psi := (u_1, \dots, u_k) \in \mathbb{R}^{|\mathcal{S}| \times k}$ and $\widehat{\Psi} := (\widehat{u}_1, \dots, \widehat{u}_k) \in \mathbb{R}^{|\mathcal{S}| \times k}$. Then,*

$$
\|\Pi_\Psi^\Phi - \Pi_{\widehat{\Psi}}^\Phi\|_\Phi < \sqrt{\frac{2\epsilon}{\lambda_{k+1} - \lambda_k}}.
$$

*Proof.* Let $y_i = \Phi^{1/2}\widehat{u}_i$ for $i = 1, \dots, k$. Then, by assumption, $y_i^\top y_j = \delta_{ij}$ for all $i, j$ and

$$
\sum_{i=1}^{k} y_i^\top \Phi^{1/2} L \Phi^{-1/2} y_i - \sum_{i=1}^{k} \lambda_i < \epsilon. \tag{23}
$$

Hence, we can apply the Graph Drawing Lemma (Lemma 3.5) with $A = \Phi^{1/2} L \Phi^{-1/2}$. Since $A$ is similar to $L$, the eigenvalues of $A$ are the eigenvalues of $L$. Moreover, the eigenvectors of $A$ are $\Phi^{1/2} u_1, \dots, \Phi^{1/2} u_{|\mathcal{S}|}$. Note that the matrix having as columns the first $k$ eigenvectors of $A$ is $X := \Phi^{1/2}\Psi$. Let $Y$ be the matrix having $y_1, \dots, y_k$ as columns and note that $Y = \Phi^{1/2}\widehat{\Psi}$. Hence, our application of the Graph Drawing Lemma establishes that

$$
\|\Pi_X - \Pi_Y\|_2 < \sqrt{\frac{2\epsilon}{\lambda_{k+1}(L) - \lambda_k(L)}}. \tag{24}
$$

Since $Y$ has orthonormal columns by construction, $\Pi_Y = YY^\top = \Phi^{1/2}\widehat{\Psi}\widehat{\Psi}^\top\Phi^{1/2} = \Phi^{1/2}\widehat{\Psi}\widehat{\Psi}^\top\Phi\Phi^{-1/2}$. Since $A$ is symmetric (cf. $\mathcal{L}$ in the proof of Lemma 3.4), the columns of $X$ are orthonormal and its projector is:

$$\Pi_X = XX^T = \Phi^{1/2}\Psi\Psi^\top\Phi^{1/2} = \Phi^{1/2}\Psi\Psi^\top\Phi\Phi^{-1/2}. \tag{25}$$

So

$$\|\Phi^{1/2}(\Psi\Psi^\top\Phi - \widehat{\Psi}\widehat{\Psi}^\top\Phi)\Phi^{-1/2}\|_2 < \sqrt{\frac{2\epsilon}{\lambda_{k+1}(L) - \lambda_k(L)}}. \tag{26}$$

However, for any matrix $\Delta$,

$$\|\Phi^{1/2}\Delta\Phi^{-1/2}\|_2 = \|\Delta\|_\Phi.$$

Finally, $\Psi\Psi^\top\Phi$ and $\widehat{\Psi}\widehat{\Psi}^\top\Phi$ are the $\Phi$-weighted least-squares operators, onto $\mathrm{span}(u_1, \ldots, u_k)$ and $\mathrm{span}(\widehat{u}_1, \ldots, \widehat{u}_k)$, respectively. Since $\Psi^\top\Phi\Psi = I$ and $\widehat{\Psi}^\top\Phi\widehat{\Psi} = I$, these projectors are:

$$\Pi_\Psi^\Phi = \Psi\Psi^\top\Phi, \qquad \Pi_{\widehat{\Psi}}^\Phi = \widehat{\Psi}\widehat{\Psi}^\top\Phi.$$

$\square$

# B. Proofs and Further Remarks of Section 4

## B.1. Examples of Misreported Laplacian

Gomez et al. (2024), citing (Wu et al., 2019), report the Laplacian for MDPs with asymmetric state graphs as (in our notation):

$$L_1 = I - \frac{P + P^\top}{2}. \tag{27}$$

This is not a Laplacian since the rows of $\frac{P+P^\top}{2}$ do not sum to one unless $P$ is doubly stochastic. This corresponds to a uniform $\phi$. Luckily, Gomez et al. (2024) mostly consider the doubly stochastic setting, as is the one induced by uniform policies on gridworlds. Moreover, they report a correct general formulation in their Appendix.

Touati et al. (2023), also citing (Wu et al., 2019), report the Laplacian as:

$$L_2 = I - \frac{P\Phi^{-1} + \Phi^{-1}P^\top}{2}. \tag{28}$$

This also is not a Laplacian because the rows of the weight matrix do not sum to one. This expression seems to be copied from (Wu et al., 2019) overlooking the fact that it is meant to be used in a weighted Hilbert space (where $L_2$ *is the expression* of a Laplacian).

In both cases, the mistake is likely without consequences. As discussed in Section 4, the GDO and its variants as commonly implemented correctly recover the representation originally defined by Wu et al. (2019). Moreover, all of these expressions are equivalent when $\phi$ is the uniform distribution, a restrictive but common assumption. However, these cases serve as examples of the care needed to handle the original formulation and how our reformulation can help avoid further confusion.

## B.2. Proofs of the Two Propositions

Let $(\mathcal{S}, \mathcal{F})$ be a complete measurable space, where $\mathcal{F}$ is a $\sigma$-algebra (omitted in the following). Let $\phi$ be a probability measure on $\mathcal{S}$ and $P$ a probability kernel such that $P(\cdot|s)$ is absolutely continuous w.r.t. $\phi$ and $\phi(s) = \int P(s|s')\mathrm{d}\phi(s')$ for all $s \in \mathcal{S}$ (all integrals are over $\mathcal{S}$). This is the familiar setting of an MDP with state space $\mathcal{S}$ and a policy $\pi$, where $P$ and $\phi$ are, respectively, the transition kernel and the stationary distribution induced by $\pi$.

Let $\mathcal{L}^2(\mathcal{S}, \phi)$ denote the set of $\phi$-square-integrable functions, that is, functions $f : \mathcal{S} \to \mathbb{R}$ such that

$$\int |f(s)|^2\mathrm{d}\phi(s) < \infty. \tag{29}$$

Let $\widetilde{\mathcal{H}}$ denote the Hilbert space obtained by equipping $\mathcal{L}^2(\mathcal{S}, \phi)$ with the following inner product:

$$\langle f, g \rangle_{\widetilde{\mathcal{H}}} = \int f(s)g(s)\mathrm{d}\phi(s). \tag{30}$$

Given any linear operator $\widetilde{A} : \widetilde{\mathcal{H}} \to \widetilde{\mathcal{H}}$, we denote the function obtained by applying $\tilde{A}$ to function $f \in \widetilde{\mathcal{H}}$ as $\widetilde{A}[f]$. Wu et al. (2019) define the Laplacian operator on $\widetilde{\mathcal{H}}$ as:

$$\widetilde{L}[f] = f - \widetilde{D}[f], \tag{31}$$

where

$$\widetilde{D}[f](s) = \int \left( \frac{1}{2} \frac{\mathrm{d}P(s'|s)}{\mathrm{d}\phi(s')} + \frac{1}{2} \frac{\mathrm{d}P(s|s')}{\mathrm{d}\phi(s)} \right) f(s') \mathrm{d}\phi(s'). \tag{32}$$

Now let $\mu$ be a $\sigma$-finite measure on $\mathcal{S}$ such that $\phi$ (hence also $P(\cdot|s)$) is absolutely continuous w.r.t $\mu$, denoted $\phi \ll \mu$ in the following. Think of $\mu$ as a reference measure such as the Lebesgue measure for continuous state spaces or the counting measure for discrete ones. Let $\mathcal{H}$ denote $\mathcal{L}^2(\mathcal{S}, \mu)$ equipped with the trivial inner product $\langle f, g \rangle_{\mathcal{H}} = \int f(s)g(s)\mathrm{d}\mu(s)$. We will denote this simply as $\langle f, g \rangle$. Also we denote the function obtained by applying a linear operator $A : \mathcal{H} \to \mathcal{H}$ to function $f \in \mathcal{H}$ simply as $Af$. This will allow us to clearly distinguish operations on $\widetilde{\mathcal{H}}$ from operations on $\mathcal{H}$ while keeping a compact operator notation. We define our Laplacian for this general setting as the unique Hilbert-Schmidt operator on $\mathcal{H}$ satisfying:[7]

$$\Phi L = \Phi - K, \tag{33}$$

where

$$Kf(s) = \int \left( \frac{1}{2} \frac{\mathrm{d}P(s'|s)}{\mathrm{d}\mu(s')} \frac{\mathrm{d}\phi(s)}{\mathrm{d}\mu(s)} + \frac{1}{2} \frac{\mathrm{d}P(s|s')}{\mathrm{d}\mu(s)} \frac{\mathrm{d}\phi(s')}{\mathrm{d}\mu(s')} \right) f(s') \mathrm{d}\mu(s'), \tag{34}$$

and

$$\Phi f(s) = \frac{\mathrm{d}\phi(s)}{\mathrm{d}\mu(s)} f(s). \tag{35}$$

To see that this is uniquely defines $L$, first note that $\ker(\Phi) = \{0\}$. Indeed, $\Phi f = \int \frac{\mathrm{d}\phi(s)}{\mathrm{d}\mu(s)} f(s) \mathrm{d}\mu(s) = 0$ only if $f = 0$ $\mu$-almost everywhere, since $\phi \ll \mu$. But then $\Phi L' = \Phi - K$ implies $\Phi(L - L') = 0$, which implies $L = L'$ since $\Phi$ is injective. Also notice that $\Phi L$ is self-adjoint since both $\Phi$ and $K$ are.

We are now ready to prove a more general version of Proposition 4.1. We remark that this equivalence is already implicit in previous work, for example (Ahmed et al., 2025).

**Lemma B.1.** *Let $f, g \in \mathcal{H} \cap \widetilde{\mathcal{H}}$. Then:*

$$\langle f, g \rangle_{\widetilde{\mathcal{H}}} = \langle f, \Phi g \rangle \qquad \qquad and \qquad \qquad \langle f, \widetilde{L}[g] \rangle_{\widetilde{\mathcal{H}}} = \langle f, \Phi L g \rangle.$$

*Proof.* The first equivalence follows from Radon-Nikodym's theorem:

$$\langle f, g \rangle_{\widetilde{\mathcal{H}}} = \int f(s)g(s)\mathrm{d}\phi(s) \tag{36}$$

$$= \int f(s)g(s)\frac{\mathrm{d}\phi(s)}{\mathrm{d}\mu(s)}\mathrm{d}\mu(s) \tag{37}$$

$$= \langle f, \Phi g \rangle. \tag{38}$$

As for the second:

$$\langle f, \widetilde{L}[g] \rangle_{\widetilde{\mathcal{H}}} = \int f(s) \left( g(s) - \widetilde{D}[g](s) \right) \mathrm{d}\phi(s) \tag{39}$$

$$= \int f(s) \left( g(s) - \widetilde{D}[g](s) \right) \frac{\mathrm{d}\phi(s)}{\mathrm{d}\mu(s)}\mathrm{d}\mu(s) \tag{40}$$

$$= \int f(s)g(s)\frac{\mathrm{d}\phi(s)}{\mathrm{d}\mu(s)}\mathrm{d}\mu(s) - \int f(s)\widetilde{D}[g](s)\frac{\mathrm{d}\phi(s)}{\mathrm{d}\mu(s)}\mathrm{d}\mu(s) \tag{41}$$

$$= \int f(s)g(s)\frac{\mathrm{d}\phi(s)}{\mathrm{d}\mu(s)}\mathrm{d}\mu(s) - \int f(s)\widetilde{D}[g](s)\frac{\mathrm{d}\phi(s)}{\mathrm{d}\mu(s)}\mathrm{d}\mu(s) \tag{42}$$

$$= \langle f, \Phi g \rangle - \int f(s)\widetilde{D}[g](s)\frac{\mathrm{d}\phi(s)}{\mathrm{d}\mu(s)}\mathrm{d}\mu(s), \tag{43}$$

---

[7]$L\Phi$ denotes the composition of $L$ and $\Phi$.

where the second equality is by Radon-Nikodym's theorem. Let us focus on the second term. By multiple applications of Fubini's and Radon-Nikodym's theorems and the chain rule of the Radon-Nikodym derivative:

$$\int f(s) \widetilde{D}[g](s) \frac{\mathrm{d}\phi(s)}{\mathrm{d}\mu(s)} \mathrm{d}\mu(s) = \int f(s) \left( \int \left( \frac{1}{2} \frac{\mathrm{d}P(s'|s)}{\mathrm{d}\phi(s')} + \frac{1}{2} \frac{\mathrm{d}P(s|s')}{\mathrm{d}\phi(s)} \right) g(s') \mathrm{d}\phi(s') \right) \frac{\mathrm{d}\phi(s)}{\mathrm{d}\mu(s)} \mathrm{d}\mu(s) \tag{44}$$

$$= \int f(s) \left( \int \left( \frac{1}{2} \frac{\mathrm{d}P(s'|s)}{\mathrm{d}\phi(s')} + \frac{1}{2} \frac{\mathrm{d}P(s|s')}{\mathrm{d}\phi(s)} \right) g(s') \frac{\mathrm{d}\phi(s')}{\mathrm{d}\mu(s')} \mathrm{d}\mu(s') \right) \frac{\mathrm{d}\phi(s)}{\mathrm{d}\mu(s)} \mathrm{d}\mu(s) \tag{45}$$

$$= a + b, \tag{46}$$

where

$$a = \frac{1}{2} \int f(s) \left( \int \frac{\mathrm{d}P(s'|s)}{\mathrm{d}\phi(s')} \frac{\mathrm{d}\phi(s')}{\mathrm{d}\mu(s')} g(s') \mathrm{d}\mu(s') \right) \frac{\mathrm{d}\phi(s)}{\mathrm{d}\mu(s)} \mathrm{d}\mu(s) \tag{47}$$

$$= \frac{1}{2} \int f(s) \left( \int \frac{\mathrm{d}P(s'|s)}{\mathrm{d}\mu(s')} g(s') \mathrm{d}\mu(s') \right) \frac{\mathrm{d}\phi(s)}{\mathrm{d}\mu(s)} \mathrm{d}\mu(s) \tag{48}$$

$$= \int f(s) \left( \int \frac{1}{2} \frac{\mathrm{d}P(s'|s)}{\mathrm{d}\mu(s')} \frac{\mathrm{d}\phi(s)}{\mathrm{d}\mu(s)} g(s') \mathrm{d}\mu(s') \right) \mathrm{d}\mu(s), \tag{49}$$

$$\tag{50}$$

while

$$b = \frac{1}{2} \int f(s) \left( \int \frac{\mathrm{d}P(s|s')}{\mathrm{d}\phi(s)} g(s') \frac{\mathrm{d}\phi(s')}{\mathrm{d}\mu(s')} \mathrm{d}\mu(s') \right) \frac{\mathrm{d}\phi(s)}{\mathrm{d}\mu(s)} \mathrm{d}\mu(s) \tag{51}$$

$$= \frac{1}{2} \int \int f(s) \frac{\mathrm{d}P(s|s')}{\mathrm{d}\phi(s)} \frac{\mathrm{d}\phi(s)}{\mathrm{d}\mu(s)} \frac{\mathrm{d}\phi(s')}{\mathrm{d}\mu(s')} g(s') \mathrm{d}\mu(s') \mathrm{d}\mu(s) \tag{52}$$

$$= \frac{1}{2} \int \int f(s) \frac{\mathrm{d}P(s|s')}{\mathrm{d}\mu(s)} \frac{\mathrm{d}\phi(s')}{\mathrm{d}\mu(s')} g(s') \mathrm{d}\mu(s') \mathrm{d}\mu(s) \tag{53}$$

$$= \int f(s) \left( \int \frac{1}{2} \frac{\mathrm{d}P(s|s')}{\mathrm{d}\mu(s)} \frac{\mathrm{d}\phi(s')}{\mathrm{d}\mu(s')} g(s') \mathrm{d}\mu(s') \right) \mathrm{d}\mu(s). \tag{54}$$

Hence:

$$a + b = \int f(s) K g(s) \mathrm{d}\mu(s) = \langle f, Kg \rangle, \tag{55}$$

which combined with Equations (43) and (46) gives[8]

$$\langle f, \widetilde{L}[g] \rangle_{\widetilde{\mathcal{H}}} = \langle f, (\Phi - K)g \rangle = \langle f, \Phi Lg \rangle, \tag{56}$$

by definition of $L$. □

**Proof of Proposition 4.1.** When $\mathcal{S}$ is finite ($|\mathcal{S}| = d$) and $\mu$ is the counting measure, $\mathcal{H}$ is just $\mathbb{R}^d$ equipped with the Euclidean dot product and the Laplacian from Equation (33) reduces to the one defined in Equation (6) since

$$K = \frac{\Phi P + P^T \Phi}{2} \qquad \Longrightarrow \qquad \Phi L := \Phi - K = \Phi - \frac{\Phi P + P^T \Phi}{2}, \tag{57}$$

hence

$$L = I - \Phi^{-1} K = I - \frac{P + \Phi^{-1} P^T \Phi}{2}, \tag{58}$$

since $\Phi$ is invertible. So Proposition 4.1 follows from Lemma B.1. □

**Proposition 4.2.** *The optimal value of Program (12) is $\sum_{i=1}^{k} \lambda_i$, where $\lambda_1, \ldots, \lambda_k$ are the $k$ smallest eigenvalues of $L$. Moreover, the solutions $x_1^*, \ldots, x_k^*$ are the first $k$ eigenvectors of $L$ according to the same ordering.*

---

[8]Since $\Phi$ is self-adjoint, this is also equal to $\langle \Phi f, Lg \rangle$.

*Proof.* With a change of variable $y_i = \Phi^{1/2}x_i$, we obtain the equivalent program:

$$\min_{x_1,\ldots,x_k} \quad \sum_{i=1}^{k} y_i^\top \Phi^{1/2} L \Phi^{-1/2} y_i \tag{59}$$
$$\text{s.t.} \quad y_i^\top y_j = \delta_{ij}, \qquad\qquad i,j \in [d].$$

Its solutions $y_1^*, \ldots, y_k^*$ are the first $k$ eigenvectors of $A = \Phi^{1/2}L\Phi^{-1/2}$ corresponding to the $k$ smallest eigenvalues. Since $A$ is similar to $L$, they have the same eigenvalues and, if $u_1, \ldots, u_d$ are the eigenvectors of $L$, the eigenvectors of $A$ are $y_i^* = \Phi^{1/2}u_i$ for $i = 1, \ldots, d$. Reverting the change of variables, the solutions of the original program are $x_i^* = \Phi^{-1/2}y_i^* = u_i$ for $i = 1, \ldots, k$. The change of variable does not change the optimal value, which in both cases is $\sum_{i=1}^{k} \lambda_i$, where $\lambda_1, \ldots, \lambda_k$ are the $k$ smallest eigenvalues of $L$ (and $A$). $\qquad\square$

## C. Auxiliary Results

**Lemma C.1** (Projector distances and principal angles). *Let $\mathcal{X}, \mathcal{Y} \subseteq \mathbb{R}^d$ be two subspaces with $\dim(\mathcal{X}) = \dim(\mathcal{Y}) = m$. Let $\Pi_\mathcal{X}$ and $\Pi_\mathcal{Y}$ be the Euclidean orthogonal projectors onto $\mathcal{X}$ and $\mathcal{Y}$, respectively, and let $\theta_1, \ldots, \theta_m$ be the principal angles between $\mathcal{X}$ and $\mathcal{Y}$. Then*

$$\|\Pi_\mathcal{X} - \Pi_\mathcal{Y}\|_F^2 = 2\sum_{i=1}^{m} \sin^2(\theta_i),$$

*or equivalently,*

$$\|\mathbf{Sin\Theta}(\mathcal{X}, \mathcal{Y})\|_F = \frac{\|\Pi_\mathcal{X} - \Pi_\mathcal{Y}\|_F}{\sqrt{2}}.$$

*Moreover,*

$$\|\Pi_\mathcal{X} - \Pi_\mathcal{Y}\|_2 = \|\mathbf{Sin\Theta}(\mathcal{X}, \mathcal{Y})\|_2 = \|(I - \Pi_\mathcal{Y})\Pi_\mathcal{X}\|_2 = \|(I - \Pi_\mathcal{X})\Pi_\mathcal{Y}\|_2.$$

*Proof.* See Edelman et al. (1998, Sec. 4.3). $\qquad\square$

## D. Visualizations

To provide a visual intuition of the role of the MDP connectivity on $\lambda_2$ we set up a 30x30 gridworld environment with an increasing number of walls. Each environment is accompanied by the corresponding induced MDP transition graph, plotted using NetworkX (Hagberg & Conway, 2020). The layout of the graph is generated using spectral graph drawing (Koren, 2003) to highlight connectivity. For each plot the value of the algebraic connectivity of the graph, $\lambda_2$ is provided. As expected, in Figure 6, 7, 8 and 9 we observe a decreasing value of $\lambda_2$ as the number of walls increases.

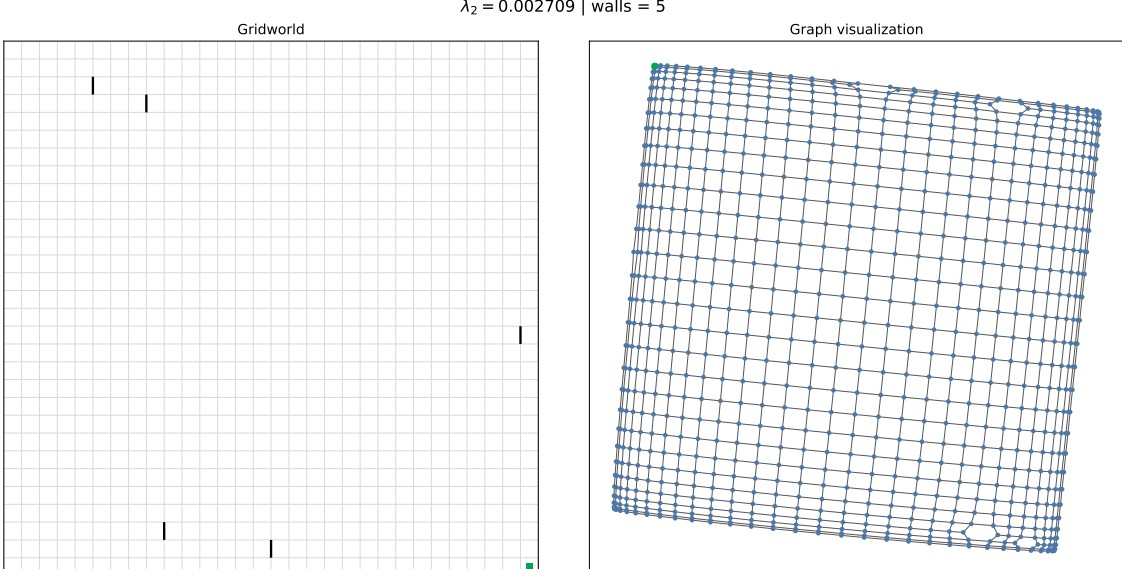

*Figure 6.* 30x30 gridworld with 5 walls with its corresponding graph visualization

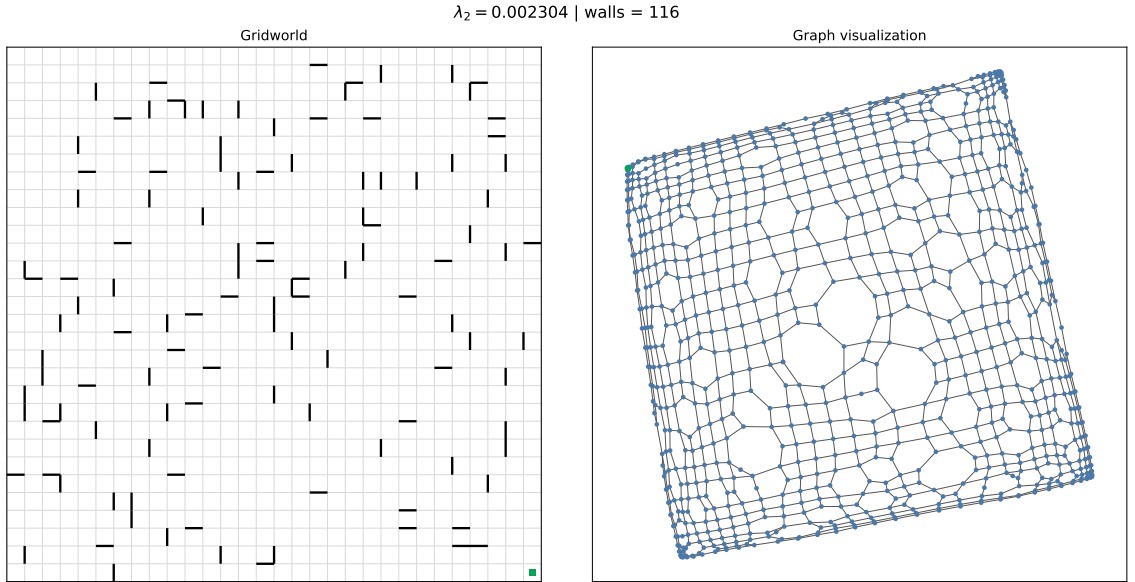

*Figure 7.* 30x30 gridworld with 116 walls with its corresponding graph visualization

$\lambda_2 = 0.001853 \mid \text{walls} = 283$

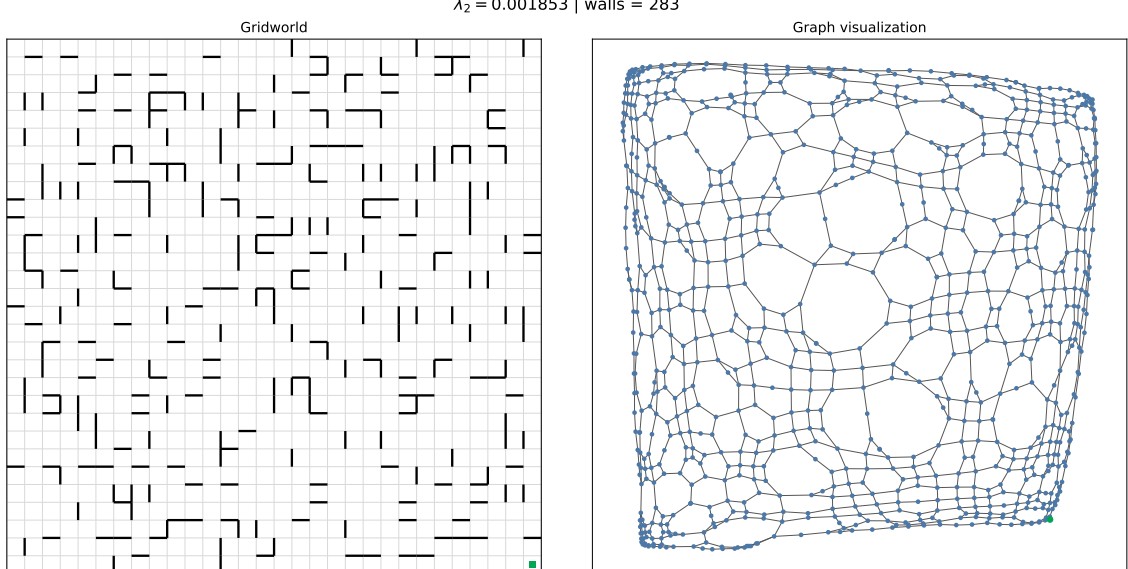

*Figure 8.* 30x30 gridworld with 283 walls with its corresponding graph visualization

$\lambda_2 = 0.001281 \mid \text{walls} = 449$

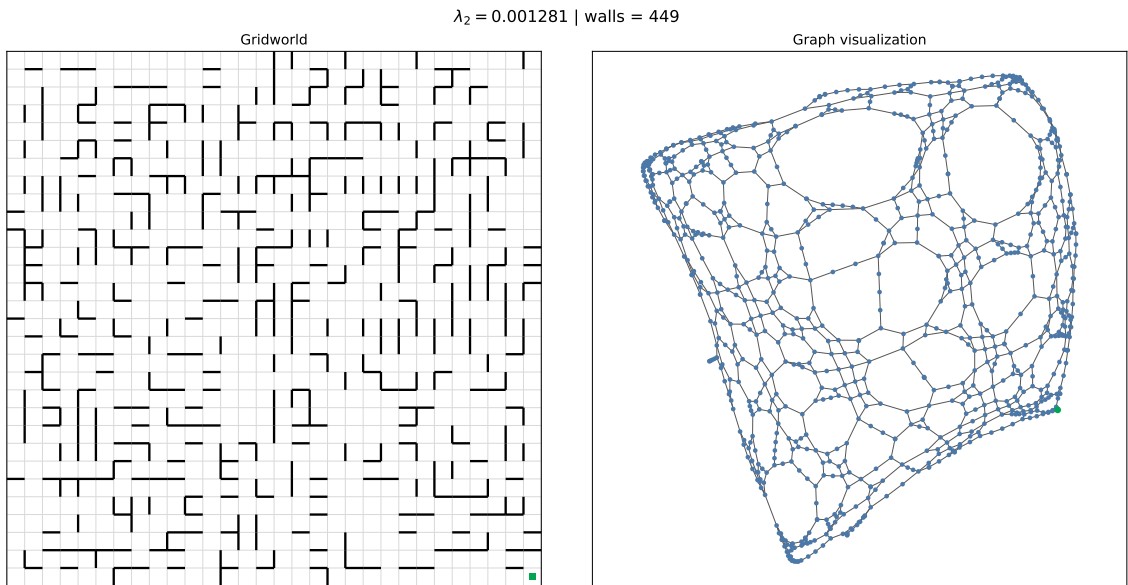

*Figure 9.* 30x30 gridworld with 449 walls with its corresponding graph visualization

## E. Additional Simulations

In this section, we report additional simulations complementing the experiments in the main text. The goal is to test whether the observed relation between graph connectivity and approximation error persists beyond the deterministic gridworld setting considered in Section 6. Overall, the additional results confirm the same qualitative behavior: environments with larger algebraic connectivity yield smaller analytical and empirical errors.

### E.1. Simulations on Stochastic Environments

We first evaluate the effect of stochastic transitions. Starting from the deterministic gridworld dynamics, we define a noisy transition kernel $P_\eta$ as follows: with probability $1 - \eta$, the transition follows the original deterministic dynamics, while with probability $\eta$, the next state is sampled uniformly at random, independently of the selected action. Thus, $\eta = 0$ recovers the

deterministic environment, whereas larger values of $\eta$ induce increasingly connected transition graphs.

Figure 10 reports the results for different values of $\eta$. As the noise level increases, the algebraic connectivity $\lambda_2$ increases, while both the analytical error $\varepsilon_{\text{ana}}$ and the GDO-based error $\varepsilon_{\text{GDO}}$ decrease. This is consistent with Theorem 3.3: better connected transition graphs lead to lower approximation errors. The variability of the error across the five GDO seeds is at least one order of magnitude smaller than the error itself, which is why the 10–90 band is not visible in the figure. For reference, Table 1 reports the corresponding values of $\lambda_2$, $\varepsilon_{\text{ana}}$, and $\varepsilon_{\text{GDO}}$.

*Table 1.* Values of $\lambda_2$, analytical error $\varepsilon_{\text{ana}}$, and GDO error $\varepsilon_{\text{GDO}}$ for different values of $\eta$.

| $\eta$ | $\lambda_2$ | $\varepsilon_{\text{ana}}$ | $\varepsilon_{\text{GDO}}$ |
|---|---|---|---|
| 0 | 0.0109 | 0.172 | $1.81 \pm 0.0244$ |
| 0.1 | 0.110 | 0.124 | $0.323 \pm 0.0155$ |
| 0.2 | 0.209 | 0.0928 | $0.187 \pm 0.00523$ |
| 0.3 | 0.308 | 0.0706 | $0.128 \pm 0.00277$ |
| 0.4 | 0.407 | 0.0538 | $0.0924 \pm 0.000831$ |
| 0.5 | 0.506 | 0.0405 | $0.0662 \pm 0.000249$ |
| 0.6 | 0.604 | 0.0296 | $0.0470 \pm 0.000413$ |
| 0.7 | 0.703 | 0.0205 | $0.0314 \pm 0.000408$ |
| 0.8 | 0.802 | 0.0127 | $0.0191 \pm 2.29 \times 10^{-5}$ |
| 0.9 | 0.901 | 0.00594 | $0.00876 \pm 2.93 \times 10^{-5}$ |

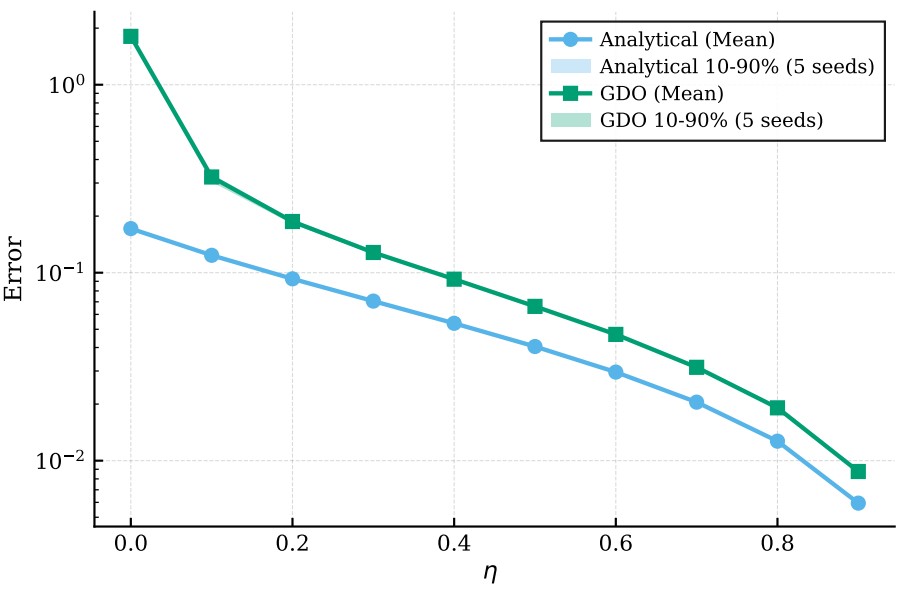

*Figure 10.* Effect of stochastic transitions on connectivity and approximation error. Increasing the stochasticity parameter $\eta$ improves the algebraic connectivity $\lambda_2$ and reduces both the analytical error $\varepsilon_{\text{ana}}$ and the GDO-based error $\varepsilon_{\text{GDO}}$.

### E.2. Simulations on Different Tasks

We also evaluate whether the same trend holds across different downstream policy-evaluation tasks. We consider a deterministic 10x10 gridworld with four actions and a uniform behavior policy. The reward is $+1$ at the goal state and $-1$ elsewhere. We vary the goal location, considering the four corners of the grid and the center, while keeping the representation dimension fixed. We perform policy-evaluation using both analytical and GDO-based representation truncating at $k = 20$ eigenvectors and compute the error as the number of walls increases. As shown in Figure 11 and 12, as the MDP connectivity reduces (higher number of walls), the approximation error calculated using $\Phi$-norm, increases for both representations.

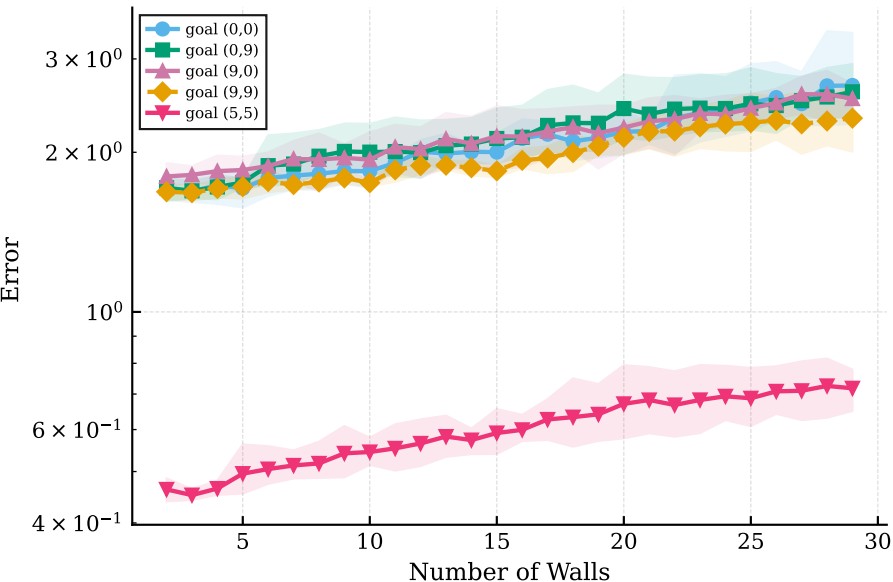

*Figure 11.* Policy-evaluation errors across different reward tasks when the representation is computed via GDO. Representation is truncated at $k = 20$, on a $10 \times 10$ grid, results averaged over 5 seeds. Each task corresponds to a different goal location. Across tasks, better connected transition graphs lead to smaller GDO-based approximation errors. In most of the tasks GDO exhibit higher error compared to analytical, but it preserves the same increasing trend as the connectivity of the graph shrinks due to walls.

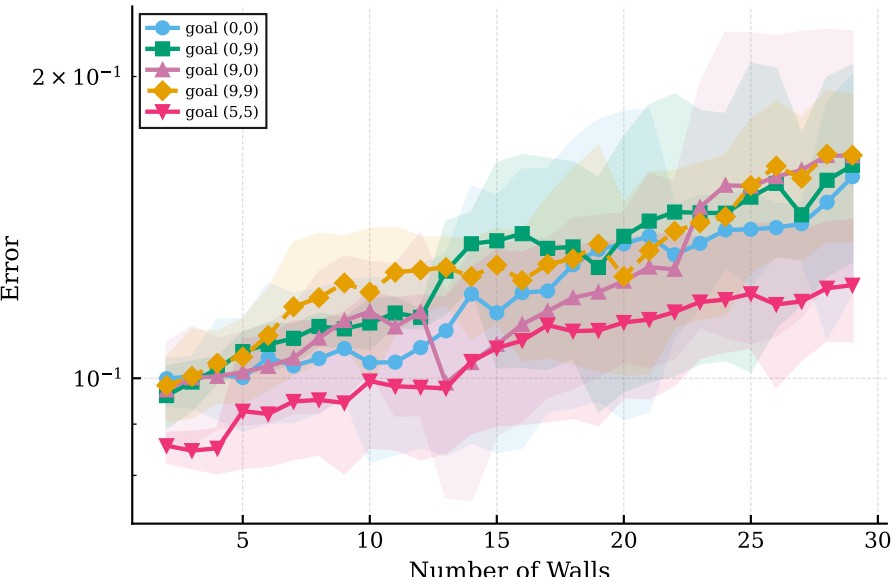

*Figure 12.* Policy-evaluation errors across different reward tasks, using the analytical representation. Representation is truncated at $k = 20$, on a $10 \times 10$ grid, results averaged over 5 seeds. Each task corresponds to a different goal location. Across tasks, better connected transition graphs lead to smaller analytical approximation errors.

