# OpenReview forum: "Impact of Connectivity on Laplacian Representations in Reinforcement Learning"
_ICML.cc/2026/Conference — ICML 2026 regular_

### Official Review · Reviewer_DWcZ · 2026-02-22

**Soundness:** 3
**Presentation:** 3
**Significance:** 3
**Originality:** 3
**Overall Recommendation:** 4
**Confidence:** 4

**Summary:**

The paper discusses linear value function approximation using the Laplacian representation as input in the average-reward reinforcement learning setting. The authors prove an upper bound for the error on the estimated value function which depends on the connectivity of the graph, the truncated eigenvectors, and the error induced by learning the Laplacian representation. Additionally, they discuss some ambiguities regarding the Laplacian definition in the Literature. Finally, the proposed method is evaluated on a grid world example.

**Compliance With Llm Reviewing Policy:**

Affirmed.

**Final Justification:**

In the rebuttal, I had one main concern regarding how prior work was framed. The authors initially had comments in their introduction and motivation that I believed did not accurately reflect existing work. They have updated their discussion, and I believe the remainder of the paper introduces some novel analysis and results connecting value function approximation and representation learning in reinforcement learning. I have updated my score from reject to weak accept.

The authors provide a novel analysis of the accuracy of the learned value function in the average-reward MDPs based on the Laplacian representation. Additionally, they discuss the existing literature's presentation of the Laplacian representation and provide their own perspective. This, while I believe is not entirely novel, is an accurate representation and provides a slightly new perspective on how learning the representation objective can be defined.

**Key Questions For Authors:**

1) All the errors in the main theorem depend on the stationary distribution, if the exploration policy used to learn the representation have low visitation for some states, doesn't their error values have a low weight in terms of the total error?

**Limitations:**

yes

**Strengths And Weaknesses:**

**Strengths**
1) The results on the relationship between the truncation error, connectivity, and error from GDO on the value function error are well presented and provide great insights.
2) To the best of my knowledge, the paper provides the first result on the relationship between the Laplacian error and the learned value function.

**Weaknesses**

A large part of the paper is discussing the ambiguities regarding the Laplacian in the literature. My main concern is regarding some of the claims about existing literature. The first 3 points are the main reason for my evaluation.
1) For example, in the introduction the authors mention "Most existing analyses implicitly assume uniform policies or symmetric transition graphs." The existing literature cited in the paper do not make either assumptions. First, all existing literature force the Laplacian operator to be self-adjoint in the same way this paper does. Second, Wu et al. (2018) derive the Laplacian representation for a generic, not necessarily uniform, policy. Additionally, while Wang et al. (2021) and Gomez et al. (2024) provide the analysis assuming a uniform policy  or a finite state-space in the main paper, they provide the general extension to any policy and state-space in the supplementary materials. Finally, there also exists work that do not assume a fixed policy, see (Wu et al. 2018, Appendix D.2.2.), (Klissarov and Machado, 2023), and (Ahmed et al. 2025).
2) The authors mention "We suspect this is why the Laplacian is incorrectly (or ambiguously) reported, for example, by Gomez et al. (2024) and Touati et al. (2023)". As I noted earlier, Gomez et al. include the general derivation in the Appendix, and their main derivation is correct given that the stationary distribution is uniform.
3) While the analysis in the Appendix on the equivalence between equation (6) and Wu et al.'s work is novel, the expression itself is not new. For the finite state-space, which the paper addresses in the main body, the equivalence is a simple substitution in the equations presented by Wu et al. Additionally, Ahmed et al. (2025) mention this equivalence in the experiments section although using different notations.
4) I would also recommend increasing the scope of experiments as a single size grid might not be sufficient to show the impact of the work. I understand that there is some varying using the walls, but introducing other grid sizes, other policies, or other settings could greatly enhance the impact.
5) I believe presenting the Lemmas in the main paper and appendix prior to using them would provide a better reading experience.


**Additional References**
- Klissarov, Martin, and Marlos C. Machado. "Deep laplacian-based options for temporally-extended exploration." arXiv preprint arXiv:2301.11181 (2023).
- Ahmed, M. H., Bhargav, J., & Ghasemi, M. (2025, October). Online Laplacian-Based Representation Learning in Reinforcement Learning. In International Conference on Machine Learning (pp. 730-751). PMLR.

---

> ### Author Rebuttal · Authors · 2026-03-30
>
> We thank the reviewer for the feedback.
>
> **General remark on the claims regarding existing literature.** We acknowledge that the tone of our claim is too strong and unnecessarily general, although we were driven by two opinions that we hold firm:
>
> (a) That of symmetry of the policy-induced transition kernel is a too strong of an assumption and
>
> (b) The results provided by [1] are very general but unfortunately easy to misunderstand and misreport because of the way they are stated.
>
> Given this, we think that our claims can be easily rephrased to be more precise and more fair to the existing literature without affecting in any way our technical contributions. In the following we provide more detailed answers that should give a precise idea on how our claims can be adjusted in the final version of the paper.
>
> > For example, in the introduction the authors mention "Most existing analyses implicitly assume uniform policies or symmetric transition graphs." The existing literature cited in the paper do not make either assumptions.
>
> > "We suspect this is why the Laplacian is incorrectly (or ambiguously) reported, for example, by [2] and [3]. As I noted earlier, [2] include the general derivation in the Appendix, and their main derivation is correct given that the stationary distribution is uniform.
>
> With "Most existing analyses implicitly assume uniform policies or symmetric transition graphs."  we were mostly referring to [2] and [3]. Namely [2] provide the following definition of the Laplacian matrix for the non-symmetric case:
>
> $$
> L_1 = I - \frac{P_{\pi} + P_{\pi}^{\top}}{2}
> $$
>
> and [3]:
>
> $$
> L_2 = I - \frac{P\Phi^{-1}+\Phi^{-1} P^\top}{2}.
> $$
>
> Those formulations lead to the same eigenvectors of the Laplacian introduced by [1] in just two cases:
>
> 1. If (as the reviewer correctly points out) the stationary distribution is uniform, that is, if $P_{\pi}$ is bistochastic, which is obtained for example obtained via an uniform behavioral policy with a deterministic environment or symmetric $P_{\pi}$ by a uniform behavioral policy with a deterministic environment or symmetric $P_{\pi}$. This is too strong of an assumption as stated in (a) above;
>
> 2. Or, in the general case, if one adopts a *$\phi$-weighted Hilbert space*, something we didn't feel like asking our readers. This is the main point of (b) above.
>
> (Weakness 3) In the "Experiments" of [4] it is stated:
> > To compute the true Laplacian representation, we perform eigendecomposition on the matrix $\hat{L}^{(t)}_{ \rho (t) }$, which is equivalent to applying the Laplacian operator in the space $\mathcal{H}(t)$
>
> So the equivalence mentioned by [4] is indeed correct. However our proposed notation, being more explicit, precisely makes remarks of this kind superfluous. Moreover, we think that the Hilbert space notation of [1] is handy for algorithm design (as the one developed in [4]), but we found it counterproductive for the kind of theoretical analysis pursued in our work. However, we will mention [4] in the final version.
>
> Indeed, the equivalence is derived from [1] and is *not* new in itself, but we felt the need to rephrase it for the sake of clarity, given the ambiguity we detected in [2] and [3]. We will make this more explicit in the paper, starting from the abstract. By "new expression" we never meant a new derivation or definition, just a clearer way to write it.
>
> (Weakness 4) We have run additional experiments varying the grid size, using a parameterized policy and an additional study on the impact of connectivity (this can be seen in the answer to reviewer iQtm). Due to space constraint we report few key results for the increased grid and different policy, a detailed description will be added in Appendix:
> (30x30 grid)
> | $\lambda_2$ | analytical error | GDO error |
> | --- | --- | --- |
> | 0.00253829 | 0.280476 | 2.04882 |
> | 0.00260298 | 0.27705 | 2.0154 |
> | 0.0026393 | 0.267125 | 1.96603 |
>
> | $k$ | mean error |
> | --- | --- |
> | 1 | 2.0124 |
> | 6 | 0.686 |
> | 11 | 0.432 |
>
> (parameterized policy)
>
> | $\lambda_2$ | analytical error | GDO error |
> | --- | --- | --- |
> | 0.0046 | 3.0811 | 50.292 |
> | 0.0068 | 0.0378 | 0.362 |
> | 0.0112 | 0.1177 | 0.662 |
>
>
> | $k$ | mean error |
> | --- | --- |
> | 1 | 9.994 |
> | 6 | 1.040 |
> | 11 | 0.636 |
>
> (Weakness 5) We thank the reviewer for the suggestion, we will enhance the presentation using the extra page
>
> (Question 1) Yes, if the visitation is low for some states, the error will place lower weight for those states. This low visitation will make $\lambda_2$ decrease though, meaning that the choice of the behavioural policy plays a crucial role for low-error theoretical guarantees.
>
> [1] Wu et al. "The laplacian in rl: Learning representations with efficient approximations"(2018)
>
> [2] Gomez et al. "Proper Laplacian representation learning"(2023)
>
> [3] Touati et al. "Does zero-shot reinforcement learning exist?"(2022)
>
> [4] Ahmed et al. "Online Laplacian-Based Representation Learning in Reinforcement Learning"(2025)

---

> > ### Author Rebuttal · Reviewer_DWcZ · 2026-03-31
> >
> > My main concerns are partially resolved. Since the manuscript can't be updated during rebuttal, I trust the authors to make the necessary changes if accepted. I kindly ask the authors to highlight some of these changes as a reply to this acknowledgement, as they just mention that they will make some changes. Additionally, in the authors' update, I encourage highlighting that while Gomez et al.'s formulation in the main paper suffers from the problems they discuss, Gomez et al. include the correct formulation in the appendix. I intend to increase my score, I am just waiting for a clear formulation of what the update is.

---

> > > ### Author Response · Authors · 2026-04-01
> > >
> > > Thank you for carefully considering our proposed improvements. Here is a detailed account of the changes we are going to make to the final version of the paper:
> > >
> > > * In the abstract: "Additionally, we provide a new expression...literature." is replaced with **"Additionally, we show how the common expression for the symmetrized MDP Laplacian is easy to misinterpret, and propose a more straightforward reformulation."** This clarifies that the definition is not novel, just stated with clearer notation.
> > > * In the Introduction: "Most existing analyses implicitly assume...dynamics." This remark is moved to the end of the Introduction, just before "Our contribution", and is changed as follows: **"In the study of Laplacian representations for RL, it is customary to assume the symmetry of the induced transition matrix for simplicity (Mahadevan & Maggioni, 2007). This assumption is satisfied, for example, by deterministic decision processes with uniform policies. However, the symmetry assumption excludes most cases of practical interest. In this work, we make no assumptions on the symmetry of the dynamics, admitting general stochastic transitions and non-uniform behavior policies. In fact, while symmetry certainly helps}, our analysis suggests that the quality of the approximation is fundamentally governed by the transition graph's connectivity"** This clarifies our positioning w.r.t. the symmetry assumption.
> > > * Towards the end of the Introduction: "In passing, we propose...literature" becomes **"Additionally, we propose a reformulation of the Laplacian for general (non-symmetric) MDPs that prevents some possible misunderstandings, of which we show some examples from the literature."**, again not to claim novelty of the symmetrized Laplacian.
> > > * In Section 4 "We suspect...(see App. B for details)" is changed into **"For example, Touati et al. (2023) and Gomez et al. (2024) report expressions for the general Laplacian that are incorrect unless making strong assumptions on the transition matrix once again (see App. B for further details). Fortunately, these oversights are harmless (these works mainly focus on the symmetric case anyway, and Gomez et al. (2024) include a correct general formulation in the Appendix), but they serve as examples of how the abstract formulation can be misleading"** to give a more fair account of the problems that affect these works and mention the correct formulation from Gomez et al.'s appendix.
> > > * We add a footnote in Section 4 just before Proposition 4.1 **"The results of this section are just a revisitation of Wu et al. 2019 under our reformulation of the Laplacian."**
> > > * In the Related Works we add **"A simplified objective was recently proposed by Ahmed et al. (2025) for online representation learning."** to the discussion of GDO refinements
> > > * In the Conclusion "Finally we clarified some ambiguities in the literature" becomes **"Finally, we proposed a more straightforward reformulation of the Laplacian for non-symmetric transition graphs"** for clarity.
> > > * In Appendix B.1 we mention **"Moreover, they report a correct general formulation in their Appendix."** when discussing Gomez et al. (2024). Please also notice how the technical details on the "misreported" Laplacians were already in this appendix. With the extra page, we could incorporate these details in Section 4.
> > > * In Appendix B.2, before proving Lemma B.1, we will mention **"We remark that this equivalence is already implicit in previous work, for example (Ahmed et al. 2025)"**
> > >
> > > Finally, we will add the **new experiments** to the paper. Given an extra page, we can include them directly in Section 6. Alternatively, we could put the new experiments in the appendix and use the extra space to mention some of the **key lemmas** from Appendix A. We would like to hear the Reviewer's opinion on this.

---

### Official Review · Reviewer_61mZ · 2026-03-05

**Soundness:** 3
**Presentation:** 3
**Significance:** 2
**Originality:** 3
**Overall Recommendation:** 3
**Confidence:** 3

**Summary:**

This paper studies the error of the Laplacian representation based on graph drawing.
Through extensive theoretical analysis, this paper shows that the total approximation error can be decomposed into two parts: the truncaton error and the feature estmation error.
The truncation error has an implicit dependency on the spectral gap $\lambda_2$, while the feature estimation error is controlled by the residual $\lambda_{k+1} - \lambda_k$.
The theoretical results indicates that environments with low connectivity may suffer from low quality of the Laplacian representation.

**Compliance With Llm Reviewing Policy:**

Affirmed.

**Key Questions For Authors:**

See "weaknesses" above.

**Limitations:**

Yes.

**Strengths And Weaknesses:**

**Strengths**
* The theoretical analysis provides a new finding that the approximation error can be decomposed into two meaningful parts.
* The paper is generally well-written and easy to follow; the notations are self-consistent and well-explained.
* The experimental results support the theoretical analysis.

**Weaknesses**
* The assumptions ($\epsilon\text{-}$optimality and linear least squares oracle) may be hard to satisfy or verify, especially for deep-model-based approximators.
* For practical environments, $\lambda_{k+1}$ may be very close or even identical to $\lambda_k$, pushing the feature estimation error to infinity and making this term meaningless.
* This paper would be much strengthened if more experimental results, particularly those of downstream RL tasks, are provided. It would be interesting to see that the theoretical findings correspond to in experiment observations.
* The authors are suggested to discuss the following papers in the Related Work section, as both of which mentioned the approximation error in Laplacian representation learning:

  [a] Wang, Kaixin, et al. "Reachability-aware Laplacian representation in reinforcement learning." Proceedings of the 40th International Conference on Machine Learning. 2023.

  [b] Fouss, Francois, et al. "Random-walk computation of similarities between nodes of a graph with application to collaborative recommendation." IEEE Transactions on knowledge and data engineering 19.3 (2007): 355-369.

---

> ### Author Rebuttal · Authors · 2026-03-30
>
> We thank the reviewer for the feedback.
>
> (1) These kinds of "oracle" assumptions, besides being common in theoretical machine learning, are just a way to focus on the aspects of the problem that we are more interested in. In our case, they allow us to "abstract away" mere issues of estimation to focus on the propagation of error due to the fundamental structure of the policy evaluation problem. Namely, $\epsilon$-optimality is our characterization of the eigenvector estimation error. Use a good estimation algorithm (e.g. GDO) and $\epsilon$ will be small.
>
> Studying the accuracy of GDO and its variants is a whole other line of work (e.g. [1],[2]). Similarly, the accuracy of linear regression is well studied in the machine learning literature [3]: assuming a least squares oracle is just a way to focus on more interesting and less understood components of the policy evaluation problem.
>
> As long as these assumptions reflect a practical real scenario (and we believe they do), they provide a flexible way of connecting our findings with other error bounds, both existing ones or future improved ones.
>
> (2)
> > For practical environments, $\lambda_{k+1}$ may be very close or even identical to $\lambda_{k}$, pushing the feature estimation error to infinity and making this term meaningless
>
> We address this concern in lines L267-L271 in the paper:
>
> > Although our result does not provide theoretical guarantees that this difference is strictly greater than 0, a recent work [4], shows that with high probability this gap is not null for the Laplacian matrix of random Erdos-Renyi graphs.
>
> For this reason we argue that the probability of a null eigengap is negligible.
> Additionally we would like to provide further information regarding this eigengap in this response to the reviewer.
>
> When the eigengap is zero, it signals the presence of an eigenvalue with multiplicity $h > 1$, which is typically associated with symmetries or structural redundancies in the graph. In that case, $h$ eigenvectors are needed to correctly describe the corresponding eigenspace.
> When the eigengap vanishes, the divergence of the bound in Lemma 3.5 should not be read primarily as a feature-approximation problem. Rather, it reflects the fact that the approximation associated with that component is incomplete: we would need dditional eigenvectors to faithfully represent the full eigenspace. In other words, truncating the spectral description at an eigenvalue whose associated symmetry is only partially captured means we lose any guarantee on the estimation error, not because the approximation is poor in quality, but because the description itself is structurally incomplete (and therefore likely incorrect). We will include this remark in the updated version of the paper.
>
> (3) We ask the reviewer to kindly clarify what it means with "those of downstream RL tasks". Regarding the additional experiments request, we have added additional experiments (please look at our answer to reviewer iQtm and reviewer DWcZ)
>
> > The authors are suggested to discuss the following papers in the Related Work section
>
> We thank the reviewer for the suggestion, we will mention these works in the Related Works
>
> [1] Gomez et al. "Proper Laplacian representation learning" (2024)
>
> [2] Wang et al. "Towards better laplacian representation in reinforcement learning with generalized graph drawing" (2021)
>
> [3] Lan, Charline Le, et al. "On the generalization of representations in reinforcement learning" (2022)
>
> [4] Christoffersen et al. "Eigenvalue gaps of the Laplacian of random graphs" (2024)

---

> > ### Author Rebuttal · Reviewer_61mZ · 2026-04-02
> >
> > Thank the authors for the response.
> > 1. For the assumptions and "eigengap > 0" claim, do you have empirical evidence showing they indeed hold in your experiments?
> > 2. For "downstream RL tasks", as stated in the introduction section, "the quality of the approximated value function for any given downstream task and policy, is fundamental for its application to policy evaluation (and possibly, optimization) with linear function approximation" (line 71 - 75), the authors may choose any task they believe is proper.

---

> > > ### Author Response · Authors · 2026-04-04
> > >
> > > We thank the reviewer for the follow-up questions, which led to useful clarifications.
> > >
> > > **Q1)** We analyzed this scenario both theoretically and empirically.
> > >
> > > As noted in the rebuttal, eigenspaces with dimension greater than 1 are typically associated with symmetries or structural redundancies in the graph. In such cases, the eigengap may be null. To address this, we extend Thm 3.3 to cover the degenerate setting and still obtain a meaningful bound.
> > >
> > > **Thm 3.3 (degenerate case, corollary)**
> > > Let $L \\in \\mathbb{R}^{|\\mathcal S|\\times |\\mathcal S|}$ be the Laplacian, with orthonormal eigenbasis $u_1,\\dots,u_{|\\mathcal S|}$ and eigenvalues
> > > $$
> > > 0=\\lambda_1 < \\lambda_2 \\le \\cdots \\le \\lambda_h < \\lambda_{h+1}=\\cdots=\\lambda_b < \\lambda_{b+1}\\le\\cdots\\leq \\lambda_{\mathcal{|S|}}
> > > $$
> > > Thus, $\\lambda_{h+1}$ has multiplicity $b-h$, with eigenspace $\\operatorname{span}(u_{h+1},\\dots,u_b)$.
> > > Let $\\widehat{u}_1,\\ldots,\\widehat{u}_k$, $\\hat v_k$, and $v$ be as in Thm 3.3. Then:
> > >
> > > $$
> > > \\|v-\\hat v_k\\|\_{\\Phi} \\le \\|\\bar r\\|\_{\\Phi}\\sqrt{\\tfrac{1}{\\lambda\_2\\lambda\_h}} + \\|v\\|_{\\Phi}\\sqrt{\\tfrac{2\\epsilon}{\\lambda\_{h+1}-\\lambda\_h}}.
> > > $$
> > >
> > > **Proof.**
> > > Using monotonicity of empirical risk under nested hypothesis classes, define
> > > $\\Psi=\\operatorname{span}(u_1,\\dots,u_k)$, $\\hat\\Psi=\\operatorname{span}(\\hat u_1,\\dots,\\hat u_k)$,
> > > $H=\\operatorname{span}(u_1,\\dots,u_h)$, $\\hat H=\\operatorname{span}(\\hat u_1,\\dots,\\hat u_h)$.
> > > Since $\\hat H \\subset \\hat\\Psi$,
> > > $$
> > > \\|v-\\Pi\_{\\hat\\Psi}^{\\Phi}v\\|\_{\\Phi} \\le \\|v-\\Pi\_{\\hat H}^{\\Phi}v\\|\_{\\Phi}.
> > > $$
> > > Applying Thm 3.3 to $\\hat H$ yields the result. $\\blacksquare$
> > >
> > > Of course using the last nonzero eigengap before $\lambda_k$ comes at a cost: the first term in the bound scales with $1/\sqrt{\lambda_h}$ instead of the smaller $1/\sqrt{\lambda_k}$. Intuitively, in this degenerate case, we unfortunately have to ignore the extra representational power that $u_{h+1},...,u_k$ may actually provide.
> > >
> > > The main reason why a null eigengap is problematic for GDO-based objectives is rooted into the formulation of GDO itself. In particular GDO generally enforces conditions such as:
> > > $$
> > > \\sum_{i=1}^k \\langle \\hat u_i, L\\hat u_i \\rangle\_{\\mathcal{H}} - \\sum_{i=1}^k \\lambda_i < \\epsilon.
> > > $$
> > > If truncation occurs inside a degenerate block ($\\lambda_k=\\cdots=\\lambda_b$), the objective cannot distinguish directions within it:
> > > $$
> > > \\sum_{i=1}^{k-1}\\lambda_i + \\lambda_k = \\sum_{i=1}^{k-1}\\lambda_i + \\lambda_j,\\quad j\\in\\{k,\\dots,b\\}.
> > > $$
> > > Thus any unit vector $v \\in \\operatorname{span}(u_k,\\dots,u_b)$ gives the same value:
> > > $$
> > > \\sum_{i=1}^{k-1}\\lambda_i + \\langle v,Lv\\rangle\_{\\mathcal{H}} = \\sum_{i=1}^{k-1}\\lambda_i + \\lambda_k.
> > > $$
> > >
> > > As a consequence, the optimizer is not unique: there exists a continuum of optimal $k$-dimensional subspaces of the form:
> > > $$
> > > \\tilde\\Psi = \\operatorname{span}(u_1,\\dots,u_{k-1},v), \\quad \\|v\\|=1.
> > > $$
> > > These can be arbitrarily far apart. Indeed, for orthogonal $v,w$:
> > > $$
> > > \\|\\Pi_{\\tilde\\Psi_v} - \\Pi_{\\tilde\\Psi_w}\\|_2 = 1.
> > > $$
> > >
> > > **Q1+Q2 (experiments)**
> > > We performed policy evaluation in a deterministic 10×10 grid (4 actions, uniform policy). Rewards are +1 at the goal and −1 otherwise. We vary the goal position (four corners of the grid and center), fix representation size $k=20$, and add up to $w=30$ walls. Results are averaged over 5 seeds.
> > >
> > > **Across walls (averaged over goals and seeds):**
> > >
> > > | $w$ | eigengap $\lambda\_{k+1}  - \lambda\_{k+1}$| $\lambda_2$ | $\varepsilon_{\text{ana}}$ | $\varepsilon_{\text{GDO}}$ |
> > > |---|---|---|---|---|
> > > | 0 | 0.0288 | 0.0245 | 0.0887 | 1.413 $\pm$ 0.0262 |
> > > | 8 | 0.0219 $\pm$ 0.00947 | 0.0211 $\pm$ 0.000882 | 0.108 $\pm$ 0.00765 | 1.601 $\pm$ 0.0578 |
> > > | 17 | 0.0205 $\pm$ 0.00913 | 0.0183 $\pm$ 0.000762 | 0.123 $\pm$ 0.00875 | 1.836 $\pm$ 0.0540 |
> > > | 25 | 0.0191 $\pm$ 0.00903 | 0.0163 $\pm$ 0.000925 | 0.145 $\pm$ 0.0110 | 2.066 $\pm$ 0.0781 |
> > > | 29 | 0.0136 $\pm$ 0.00431 | 0.0151 $\pm$ 0.00107 | 0.156 $\pm$ 0.00909 | 2.167 $\pm$ 0.159 |
> > >
> > > **Per-goal (averaged over walls and seeds):**
> > >
> > > | Goal | $\varepsilon_{\text{ana}}$ | $\varepsilon_{\text{GDO}}$ |
> > > |---|---|---|
> > > | (0,0) | 0.120 $\pm$ 0.0307 | 2.067 $\pm$ 0.354 |
> > > | (0,9) | 0.127 $\pm$ 0.0433 | 2.128 $\pm$ 0.442 |
> > > | (9,0) | 0.121 $\pm$ 0.0391 | 2.122 $\pm$ 0.419 |
> > > | (9,9) | 0.128 $\pm$ 0.0359 | 1.955 $\pm$ 0.347 |
> > > | (5,5) | 0.104 $\pm$ 0.0281 | 0.590 $\pm$ 0.159 |
> > >
> > > **Eigengap distribution, 150 unique samples (changing the goal position has no effect on the eigengap):**
> > > min 0.0056, mean 0.0231, std 0.0097.
> > >
> > > These results support our claims: as the graph becomes more connected, both analytical and GDO errors decrease regardless of the task. We also never observed a zero eigengap in practice.
> > >
> > > We will include both the degenerate-case extension and these results in the final manuscript.

---

### Official Review · Reviewer_ShVC · 2026-03-12

**Soundness:** 3
**Presentation:** 4
**Significance:** 3
**Originality:** 4
**Overall Recommendation:** 5
**Confidence:** 3

**Summary:**

This paper provides theoretical bounds and justification of using the Graph Laplacian eigenvectors for linear value function approximation in an average-reward RL setting. The proposed bound measures the value function approximation error via two terms: truncation error and feature estimation error. The first term relates the value approximation error with the spectral gap and the representation dimension. The second term captures the error induced when approximating the eigenvectors. The theory is supported with experiments in gridworlds.

**Compliance With Llm Reviewing Policy:**

Affirmed.

**Final Justification:**

The authors have addressed all of the (minor) concerns I raised. The work is highly relevant from a theoretical perspective, particularly for the representation learning community. While additional results on more complex environments could further strengthen the paper, I recognize that this may be beyond the current scope. With that said, I will be maintaining my score.

**Key Questions For Authors:**

1. The GDO objective (Eqn 5) enforces the eigenvectors to be $\Phi$-orthonormal. But in Assumption 3.1, they are assumed to be euclidean normal. Would this mean, for Lemma 3.6 to be true, Assumption 3.1 should be updated to be  $\Phi$-orthonormal?

**Limitations:**

yes

**Strengths And Weaknesses:**

**Strengths**
- The paper is well written and easy to follow, with a good level of detail.
- This work is very relevant to the current unsupervised RL community.

**Weakness**

- The experiments are currently limited to gridworld environments. It would be a stronger paper with results, similar to Figure 3, in more complex domains, such as MuJoCo.


**Minor Comments**

1. A figure illustrating the different gridworld environments used (in the Appendix?) could provide additional intuition. Similarly, a figure of the Laplacian representation might help readers better understand how they capture the structure of the MDP.

2. Figures could be rearranged to follow the same order in which they are referenced in the text. And Figure 4 currently does not appear to be mentioned in the text.

3. The  Laplacian eigenvectors have also been used for exploration in MDPs, not only in bandit settings. It may be worth citing this line of work. [1]


---


[1] Klissarov, Martin, and Marlos C. Machado. "Deep Laplacian-based options for temporally-extended exploration." ICML 2023.

---

> ### Author Rebuttal · Authors · 2026-03-30
>
> (1) We thank the reveiwer for pointing out the typo. For Lemma 3.6 to be true, the statement of Assumption 3.1 should be updated to be $\Phi$-orthonormal. We note that this updated assumption is coherent with what the GDO enforces in practice when samples are drawn from a buffer populated with samples distributed according to the stationary distribution $\phi$.
> Throughout the paper, for instance in the proof of Lemma 3.6 (L1024-L1059), we did use the correct assumption:
> > Let $y_i = \Phi^{\frac{1}{2}} \hat{u_{i}}$  for $ i = 1, \ldots,k$. Then, by assumption, $y_i^{\top}y_j = \delta_{ij}.$
>
> *Minor remarks:*
>
> We agree that your suggestions would improve the quality of the presentation. We will mention Figure 4 in the paper, re-arrange the figures and add some visual aids either to the appendix  or extra page granted to accepted papers.
>
> > The Laplacian eigenvectors have also been used for exploration in MDPs, not only in bandit settings. It may be worth citing this line of work. [1]
>
> We thank the reviewer for the suggestion, we will mention [1] in the paper.
>
> [1] Klissarov, Martin, and Marlos C. Machado. "Deep Laplacian-based options for temporally-extended exploration." ICML 2023.

---

> > ### Author Rebuttal · Reviewer_ShVC · 2026-04-03
> >
> > My minor concerns have been addressed. I maintain my score.

---

### Official Review · Reviewer_iQtm · 2026-03-21

**Soundness:** 3
**Presentation:** 2
**Significance:** 2
**Originality:** 3
**Overall Recommendation:** 3
**Confidence:** 3

**Summary:**

The paper is largely well written and makes a great case for their Graph Laplacian and how it connects to learning in an MDP. They finally give an example of environments with varying numbers of walls as a practical application of their work.

**Compliance With Llm Reviewing Policy:**

Affirmed.

**Key Questions For Authors:**

See above.

**Limitations:**

See above.

**Strengths And Weaknesses:**

# Strengths:

The paper is largely well written and makes a great case for their Graph Laplacian and how it connects to learning in an MDP. They finally give an example of environments with varying numbers of walls as a practical application of their work.

# Weaknesses:

I see two primary weaknesses:

1. While the work is well written as the results are being introduced there is not sufficient clarity in terms of what these results mean in MDP terms, intuitively. For example, I get that theorem 3.2 tells us about how the optimality gap is a function of eigenvalues of the Graph Laplacian but what does it mean intuitively in terms of MDPs? Which MDPs are harder to value estimates for (even in the presence of the oracle) vs which are not? I guess the experimental section is trying to make this connection but it remains elusive to me. Moreover, the paper is motivated as connecting the “MDP topology” to the approximation error but the idea of MDP topology is not properly intuited.

2. The experimental section only shows results for deterministic environments with walls. First, a visual representation of this environment would help me as a reader. Second, it doesn't demonstrate how much of a difference does adding stochasticity make to this connection between the GDO error and the error. I would be curious about both these issues. Moreover, I believe the experimental section could connect to the theoretical results a lot better.


# Minor issues:

Line 100 eqn on right side: I would explain how you are using the transition kernel as an operator $\mathcal P^{\pi}$.

Line 103 to 104 right side: I would cite [1] for the normalization (I think?)

Line 119-1220 left side: $\mathcal L$ should be $L$?

The paragraph on Graph Laplacian: I believe you should have introduced or defined the idea of graph connectivity or connected components by this point of the paper. Although I have an idea what it is I believe it would be beneficial for the broader community. The rest of the terms: Markov chains, ergodic, stationary distribution etc. might be accessible.

Assumption 3.2: “is the matrix having” -> “in the matrix with”, also could you specify the dimensions of $\Pi^{\Phi}_X$ please?

Line 283 left side: just a suggestion “more usable” -> “more applicable”

Figure 4 and 5: why is the relation of the error to the second eigenvalue not the same as the inverse of square root as the theory suggests?

------------
## References

[1] Policy gradient methods for reinforcement learning with function approximation, Richard S. Sutton, David A. McAllester, Satinder P. Singh, Yishay Mansour

---

> ### Author Rebuttal · Authors · 2026-03-30
>
> We thank the reviewer for the feedback.
>
> (1) To clarify the connection with MDP topology, we interpret “connectivity” as probability mass flow in the MDP graph, governed by the dynamics $P$ and behavioral policy $\pi_b$. For an MDP with states $S =$ {$S_0, \dots, S_N$}, connectivity is maximal if any state can be reached from any other in one step under $P$ and $\pi_b$. Conversely, if $\pi_b$ assigns zero probability to actions leading to certain states, parts of the state space can become poorly connected.
>
> Under this view, well-connected MDPs are better suited for Laplacian representations. Our results formalize this via the eigenvalue $\lambda_2$, which can be estimated from data (e.g using GDO-like objectives) and used to bound approximation error. The link between connectivity and Laplacian spectrum is well established in graph theory (see Sec. 2).
>
> A second intuition concerns the number of eigenvectors $k$. The approximation error from truncating to $k$ eigenvectors can be estimated, suggesting an iterative procedure that increases $k$ until a target error is reached. This is captured by the $\frac{1}{\sqrt{\lambda_k}}$ term in Theorem 3.3.
>
> Additionally $\lambda_k$ is related to the *k-way expansion constant* of the graph which measures the cost of the best partition of the graph into $k$ pieces via a Cheeger-like inequality [1]. A small value of $\lambda_k$ means the graph can be cheaply cut into $k$ well-separated groups, i.e it has $k$ approximately disconnected clusters. The eigengap $\lambda_{k+1}- \lambda_k$ tells you how "cleanly" the graph decomposes into exactly $k$ clusters vs $k+1$. We will expand the discussion in the paper in the appendix or the additional page with these intuitive remarks.
>
> (2) We initially reported deterministic experiments for simplicity, as $\pi_b$ already introduces stochasticity. Following the reviewer’s suggestion, we added experiments with stochastic transitions $P_\eta$, controlled by noise parameter $\eta$:
>
> With probability $1-\eta$, transitions follow the original deterministic dynamics.
> With probability $\eta$, the next state is sampled uniformly at random, independent of the action.
>
> Thus, $\eta=0$ recovers the deterministic case, while $\eta=1$ yields a fully connected graph. We evaluate $\eta \in $ {$0.1, \dots, 0.9$}.
>
> Results (15×15 grid, $k=20$, averaged over 5 seeds) show that as $\eta$ increases, $\lambda_2$ increases and both analytical error $\varepsilon_{\mathrm{ana}}$ and GDO error $\varepsilon_{\mathrm{GDO}}$ decrease, consistent with our theory. Full plots and discussion will be added to the appendix.
>
> | $\eta$ | $\lambda_2$ | $\varepsilon_{\mathrm{ana}}$ | $\varepsilon_{\mathrm{GDO}}$ |
> |---|---:|---:|---:|
> | $0$ | $0.01093$ | $0.1716$ | $1.858 \pm 0.01811$ |
> | $0.1$ | $0.1098$ | $0.1238$ | $0.3247 \pm 0.01156$ |
> | $0.2$ | $0.2087$ | $0.09277$ | $0.1885 \pm 0.004254$ |
> | $0.3$ | $0.3076$ | $0.07063$ | $0.1281 \pm 0.002489$ |
> | $0.4$ | $0.4066$ | $0.05381$ | $0.09219 \pm 0.002065$ |
> | $0.5$ | $0.5055$ | $0.04049$ | $0.06494 \pm 0.0009348$ |
> | $0.6$ | $0.6044$ | $0.0296$ | $0.04623 \pm 0.001123$ |
> | $0.7$ | $0.7033$ | $0.02048$ | $0.03152 \pm 0.0001987$ |
> | $0.8$ | $0.8022$ | $0.0127$ | $0.01909 \pm 4.402e-05$ |
> | $0.9$ | $0.9011$ | $0.005944$ | $0.008778 \pm 4.122e-06$ |
>
>
> > Line 100 eqn on right side: I would explain how you are using the transition kernel as an operator $\mathcal{P}^{\pi}$
>
> $\mathcal{P}^{\pi}$ is an $\mathcal{S} \times \mathcal{S}$ matrix so operations like $\mathcal{P}^{\pi} v_{\pi}$ refer to matrix-vector multiplication.
>
> > Line 103 to 104 right side: I would cite [2] for the normalization
>
> We thank the reviewer for the suggestion, the work will be cited in the final paper.
>
> > Line 119-1220 left side: $\mathcal{L}$ should be $L$?
>
> Yes, the sentence (L119-L122) was actually meant for $L$ even though the same propery holds for $\mathcal{L}$ as well.
>
> > I believe you should have introduced or defined the idea of graph connectivity or connected components by this point of the paper.
>
> We agree with the reviewer and those definitions will be added using the extra page.
>
> > could you specify the dimensions of $\Pi_{X}^{\Phi}$ please?
>
> $\Pi_{X}^{\Phi}$ is the $\Phi$-orthogonal projection matrix on to the columns of matrix $X \in \mathbb{R}^{\mathcal{S} \times k}$  so $\Pi_{X}^{\Phi} = X(X^{\top} \Phi X)^{-1}X^{\top}$ has dimension $\mathcal{S} \times \mathcal{S}$. $k \in $ {$1, \ldots, N $} is the number of eigenvectors chosen for the representation.
>
> > Figure 4 and 5: why is the relation of the error to the second eigenvalue not the same as the inverse of square root as the theory suggests?
>
> We argue that the figure shows an inverse square root dependency of the errors with respect to $\lambda_2$ in Figures 4 and 5, as the theory suggests.
>
> [1] Multiway spectral partitioning and higher-order cheeger inequalities,. Lee et al.
>
> [2] Policy gradient methods for reinforcement learning with function approximation, Sutton et al.

---

> > ### Author Rebuttal · Reviewer_iQtm · 2026-04-05
> >
> > While my concern about experiments with stochastic transitions have been addressed but my larger point about the topology of the MDP and visuals related to the experimental setup remain unresolved. I will retain my score.

---

> > > ### Author Response · Authors · 2026-04-07
> > >
> > > In the following we address the remaining concerns of the Reviewer, whom we kindly invite to reconsider the overall recommendation in light of the proposed  additions.
> > >
> > > We agree that a visualization of the MDP to better capture the link between connectivity and the experimental setup can enhance the presentation of the paper. In the following we provide a visual representation of the 30x30 gridworld environments with increasing number of walls used in the original experiments. Each environment is accompanied by the corresponding induced MDP transition graph, plotted using NetworkX [1].
> > >
> > > The layout of the graph is generated using spectral drawing [2] to better display connectivity in a visually intuitive way. For each plot the value of $\lambda_2$ (the algebraic connectivity of the graph) is provided. Coherently with our original experiments, we observe a decreasing value of $\lambda_2$ as the number of walls increases.
> > > This shows, at a glance, the connection between the structure of the MDP, the topology of the graph, and the algebraic notions of connectivity thoroughly discussed in the paper.
> > > We will incorporate these plots in the final manuscript.
> > >
> > > The plots are in the pdf file at the following (anonymous) link: https://anonymous.4open.science/r/MDP_connectivity_plot-0DFF/MDP_connectivity_plot.pdf
> > >
> > > [1] https://github.com/networkx/networkx
> > >
> > > [2] Koren, Yehuda. "On spectral graph drawing." (2003)

---

### Decision · Program_Chairs · 2026-04-30

**Decision:**

Accept (regular)

**Comment:**

This paper presents a theoretical analysis of the approximation error associated with linear value function approximation using Laplacian representations in reinforcement learning, decomposing the error into truncation and feature estimation components while linking it to the environment's spectral gap. During the review process, the reviewers recognized the theoretical contributions but raised valid concerns regarding the intuitive connection to MDP topology, the handling of zero eigengaps, the limited empirical evaluation, and the characterization of prior literature. The authors submitted a detailed rebuttal that systematically addressed these issues by introducing a corollary for degenerate eigengaps, expanding the experiments to include stochastic transitions and a downstream policy evaluation task, providing visual graph layouts, and revising the discussion of existing work. While two reviewers maintained their initial scores due to remaining reservations about the empirical scale and intuitive clarity, I reviewed the author responses and the updated evidence, determining that the core technical criticisms have been adequately resolved. The manuscript is technically sound, clearly written, and provides a useful theoretical perspective on representation learning, supporting a recommendation for acceptance.